



# Calibrating a long-term meteoric $^{10}$Be delivery rate into Western US glacial deposits through a comparison of complimentary meteoric and in situ-produced $^{10}$Be depth profiles

Travis Clow[1], Jane K. Willenbring[1,2], Mirjam Schaller[3], Joel D. Blum[4], Marcus Christl[5],

Peter W. Kubik[5], Friedhelm von Blankenburg[2]

[1]Scripps Institution of Oceanography, University of California, San Diego, La Jolla, California 92037, USA

[2]GFZ German Research Centre for Geosciences, Earth Surface Geochemistry, Telegrafenberg, 14473 Potsdam, Germany

[3]Geodynamics, University of Tübingen, Wilhelmstraße 56, 72076 Tübingen, Germany

[4]Department of Earth and Environmental Sciences, University of Michigan, Ann Arbor, MI, USA

[5]ETH Zurich, Laboratory of Ion Beam Physics, HPK G23, Schafmattstrasse 20, ETH-Zurich, CH-8093 Zurich, Switzerland

*Correspondence to*: Travis Clow (tclow@ucsd.edu)

**Abstract.** Meteoric $^{10}$Be ($^{10}$Be$_{met}$) concentrations in soil profiles have great potential as a geochronometer and a tracer of Earth

surface processes, particularly in fine-grained soils lacking quartz that would preclude the use of *in situ*-produced $^{10}$Be ($^{10}$Be$_{in\ situ}$). One prerequisite for using this technique for accurately calculating rates and dates is constraining the delivery, or flux, of $^{10}$Be$_{met}$ to a site.  However, few studies to date have quantified long-term (i.e. millennial) delivery rates. In this study, we compared existing concentrations of $^{10}$Be$_{in\ situ}$ with new measurements of $^{10}$Be$_{met}$ in soils sampled from the same depth profiles to calibrate a long-term $^{10}$Be$_{met}$ delivery rate following a newly developed methodology. We did so on the Pinedale and Bull

Lake glacial moraines at Fremont Lake, Wyoming (USA) where age, grain sizes, weathering indices, and soil properties are known, as are erosion/denudation rates calculated from $^{10}$Be$_{in\ situ}$. After ensuring sufficient beryllium retention in each profile, solving for the delivery rate of $^{10}$Be$_{met}$ via Monte Carlo simulations, and normalizing to Holocene-average paleomagnetic intensity, we calculate best-fit fluxes of 0.92 (+/- 0.08) x $10^6$ and 0.71 (+0.09/-0.08) x $10^6$ atoms cm$^{-2}$ yr$^{-1}$ to the Pinedale and Bull Lake moraines, respectively, and compare these values to two widely-used $^{10}$Be$_{met}$ delivery rate estimation methods.

Accurately estimating $^{10}$Be$_{met}$ flux using these methods requires careful consideration of spatial scale as well as temporally varying parameters (e.g. paleomagnetic field intensity) to ensure the most realistic estimates of $^{10}$Be$_{met}$-derived erosion rates in future studies.



# 1 Introduction

$^{10}$Be is a cosmogenic isotope with a half-life of 1.39 Myr (Chmeleff et al., 2010) and its meteoric form ($^{10}$Be$_{met}$) is produced in the atmosphere through spallation reactions as high-energy cosmic rays collide with target atoms in the atmosphere (Lal and Peters, 1967). $^{10}$Be$_{met}$ is then delivered to Earth's surface via precipitation or as dry deposition at a flux of $0.1 - 2 \times 10^6$ atoms cm$^{-2}$ yr$^{-1}$ followed by dissolved export in runoff, or depending on retentivity, adsorption onto fine-grained, reactive surfaces, typically clays and Fe-oxyhydroxides, in soil horizons at the Earth's surface (Willenbring and von Blanckenburg,

2010). $^{10}$Be$_{met}$ has been used as a tracer of Earth surface processes, including estimating erosion rates at the soil profile and river catchment scales, soil residence times, ages of landforms over millennial to million-year timescales, and paleo-denudation rates from marine sedimentary records (Pavich et al., 1986; McKean et al., 1993; Jungers et al., 2009; Willenbring and von Blanckenburg, 2010; von Blanckenburg et al., 2012; von Blanckenburg and Bouchez, 2014; Wittman et al., 2015; von Blanckenburg et al., 2015; Portenga et al., 2019; Jelinski et al., 2019). Prerequisites to interpreting the concentrations and

isotope ratios (i.e. $^{10}$Be$_{met}$/$^9$Be) as erosion or denudation (the sum of erosion and weathering) rates, respectively, include knowing the delivery rate of $^{10}$Be$_{met}$ (Pavich et al., 1986; Reusser et al., 2010; Graly et al, 2011; Heikkilä and von Blanckenburg, 2015; Dixon et al., 2018) and quantifying the mobility or retention of beryllium in soils (e.g. Bacon et al., 2012; Boschi and Willenbring, 2016a,b; Maher and von Blanckenburg, 2016; Dixon et al., 2018), neither of which are comprehensively evaluated in the majority of studies. The potential ability of using $^{10}$Be$_{met}$ depth profiles to obtain quantitative data on soil ages,

residence times, production- and denudation rates in a similar manner as *in situ*-produced $^{10}$Be ($^{10}$Be$_{in\ situ}$) depth profiles could prove to be highly advantageous, as it is easier to measure (due to much higher concentrations than $^{10}$Be$_{in\ situ}$) and can be employed in a much wider range of environments, as there is no dependence on the existence of coarse-grained quartz as is required for the analysis of $^{10}$Be$_{in\ situ}$. $^{10}$Be$_{met}$ shares a cosmic ray origin with ($^{10}$Be$_{in\ situ}$) which is produced within crystal lattices in surface rocks and soil, rather than in the atmosphere, with a well constrained total production rate of 4.01 atoms g$^{-1}$ yr$^{-1}$ at

sea level, high latitude (Borchers et al., 2016), and is characterized by full retentivity and known production pathways with depth. $^{10}$Be$_{met}$, in stark contrast, is potentially subjected to variable adsorption depths, incomplete retentivity, and heterogeneous internal redistribution.

In this study, we compare the previously published $^{10}Be_{in\ situ}$ depth profiles of the Pinedale and Bull Lake terminal glacial

moraines in Wind River, Wyoming (Schaller et al., 2009a,b) with new $^{10}Be_{met}$ concentrations from depth profiles from the

same sample material to evaluate the long-term (i.e. millennial) delivery rate of $^{10}Be_{met}$ to this site. This is the first study that

evaluates the $^{10}Be_{met}$ flux for eroding soils as derived from the comparison of $^{10}Be_{in\ situ}$ and $^{10}Be_{met}$ depth profiles and erosion

rates. We utilize *a priori* knowledge of effective erosion rates from Schaller et al. (2009a) to constrain and locally calibrate

the flux of $^{10}Be_{met}$ flux to these moraines while considering the extent of $^{10}Be_{met}$ retention post-delivery. We then compare the

resulting back-calculated fluxes, determined via Monte Carlo simulations, with the predicted fluxes of Graly et al. (2011) and

Heikkilä and von Blanckenburg (2015), normalizing each result for paleomagnetic field intensity variations over the Holocene.

We also explore the practical differences between these flux estimates and advocate for each approach to be carried out when

estimating $^{10}Be_{met}$ flux for use in erosion rate calculations in future studies.

## 2 Background

### 2.1 Study Area

The Fremont Lake area of the Wind River Mountains (Wyoming, United States) experienced multiple glacial advances during

the Pleistocene, evidenced by several moraines of Pinedale and Bull Lake age (Fig. 1; modified from original mapping and

descriptions by Richmond, 1987). The climate is cold, semi-arid, and windy, with a 50 year annual precipitation rate and

temperature of 0.276 m and 2.1° C, respectively (WRCC, 2005) in the proximal town of Pinedale, Wyoming (~3.5 km

southwest of the field area).

The Pinedale and Bull Lake-age terminal moraines (hereafter referred to as Pinedale and Bull Lake moraines) analyzed in this

study (Fig. 1) were formed by highland-to-valley mountain glaciers draining an ice cap accumulation zone that covered the

mountain range. The Pinedale moraine is more steep-sided and boulder-strewn than the gently sloping Bull Lake moraine,

each with a total height of ~30 m (see Figs. 1b, 1c of Schaller et al., 2009a for detailed moraine transects). The pH of the

moraine soils is well characterized; both profiles have pedogenic carbonate below 1 m, fixing the pH at depth to ~ 8 (Chadwick

and Chorover, 2001). Hall and Shroba (1995) report pH data on profiles adjacent to those analyzed in this study, with average pH ranging from ~5.5 on the surface to ~8 at depth.

The depth profile samples analyzed for $^{10}Be_{met}$ reported here are the same sample material analyzed for $^{10}Be_{in\ situ}$ by Schaller et al. (2009a). We utilize bulk samples sieved to <2 mm for our analysis, extracted from the lower mineral soil developed on each moraine, both mixtures of Archean granite, granodiorite, and dioritic gneiss. As such, the same reported depths and grain size distributions apply for each sample at depth. The primary mineral content in the deepest (unweathered) sample is (in order of decreasing abundance): plagioclase, quartz, biotite, K-feldspar, hornblende, and magnetite (Taylor and Blum, 1995).

Secondary clay minerals in the 2 μm size fraction include kaolinite, vermiculite, illite, and smectite (Mahaney and Halvorson, 1986), with total clay content ranging from 3 to 10 wt% and 9 to 30 wt% for the Pinedale and Bull Lake profiles, respectively. Major element data is reported in Schaller et al. (2009b). Sr isotope measurements of the moraine soils and dust sources showed insignificant eolian fluxes in the depth profiles of the Pinedale and Bull Lake moraines (Blum and Erel, 1997; Taylor and Blum, 1997).

**2.2 Previous Age Constraints**

Ages for each moraine have been independently determined via multiple methods, with $^{10}Be_{in\ situ}$ surface exposure ages of boulders combined with $^{230}Th/U$ ages of proximal contemporaneous fluvial terraces yielding the most reliable average estimates of 21 kyr and 140 kyr for the Type-Pinedale and Bull Lake-age moraines, respectively (Gosse et al., 1995; Phillips et al., 1997; Easterbrook et al., 2003; Sharp et al., 2003). These ages closely correspond with global maximum ice volumes of

marine oxygen isotope stages 2 and 6, respectively (Sharp et al., 2003). We recalculated the $^{10}Be$ boulder surface exposure ages used to constrain the timing of advancement of each moraine to its terminal position based on a recent revision of the $^{10}Be$ half-life, which affected the AMS standard values (Chmeleff et al., 2010), and the most recent nucleonic production rate of 3.92 atoms $g^{-1}\ a^{-1}$ at sea level-high latitude (Borchers et al., 2016) (Table 2); the updated independent age constraints are 25 kyr for the Pinedale moraine and remain at 140 kyr for the Bull Lake moraine (see Supplementary Material for details).




## 2.3 Previous Denudation Constraints

All moraine surfaces are likely to have been eroded to some extent after their deposition. To estimate the amount of erosion, we compiled denudation rates (comprising erosion and chemical loss by dissolution) for contiguous western US Pinedale and Bull Lake-aged moraines. Denudation rates have been estimated via multiple techniques, including field observations of

removed material (Meierding, 1984; Easterbrook et al., 2003), diffusion models of moraine surface lowering (Hallet and Putkonen, 1994; Putkonen and Swanson, 2003), exhumation modeling of surface exposure ages of boulders (Phillips et al., 2009), and cosmogenic depth profile dating (Phillips et al., 2000), showing considerable range (5 - 700 mm kyr$^{-1}$), likely owing to each studies' methodological assumptions and limitations, and the respective sites stability as set by geographic location. In particular, denudation rates from diffusion models of surface degradation are over an order of magnitude higher (280 - 700

mm kyr$^{-1}$) (Putkonen and Swanson, 2003) than rates determined from $^{10}Be_{in\ situ}$ (5 - 60 mm kyr$^{-1}$) (Schaller et al., 2009a).

The reported denudation rates of Schaller et al. (2009a) were calculated using a sea level, high latitude production rate of 5.1 atoms $g_{(qtz)}^{-1}$ yr$^{-1}$ (Stone, 2000) and a decay constant of 4.62 x 10$^{-7}$ y$^{-1}$. Denudation rates were recalculated using CRONUS v.3 (Phillips et al., 2016) with the updated half-life and production rate values (Table 2) and updated independent age constraints

(See Supplementary Material) scaled to the sample altitude and latitude (Dunai, 2000) assuming two denudation rate scenarios: one of constant denudation since moraine deposition, and the other of transient denudation decreasing in magnitude since moraine deposition. Recalculated denudation rates are 32.1 ± 2.7 mm kyr$^{-1}$ and 12.4 ± 4.8 mm kyr$^{-1}$ for the Pinedale and Bull Lake moraines, respectively, in the case of transient denudation, and are 15 mm kyr$^{-1}$ and 7.5 mm kyr$^{-1}$ for the Pinedale and Bull Lake moraines, respectively, in the case of constant denudation (Table 3). This results in recalculated average denudation

rates, taken to be the average value between the different denudation rate scenarios, of 23.5 mm kyr$^{-1}$ and 10 mm kyr$^{-1}$ for the Pinedale and Bull Lake moraines, respectively (Table 3) These recalculated denudation rates are determined from the best-fit Chi-Square solutions obtained from running Models 2, 4, 6, and 8 of Schaller et al. (2009a) with present-day parameters (Table 2) (See *Supplementary Material* for details). To properly compare the denudation rates of Schaller et al. (2009a) with the $^{10}Be_{met}$–derived erosion rates using the methods of von Blanckenburg et al. (2012), the weathering component of denudation



must be accounted for. For the Pinedale moraine, chemical weathering mass loss is estimated to be 16% of the denudation rate, while for the Bull Lake moraine, the chemical weathering mass loss accounts for 20% (Schaller et al., 2009b). If the weathering mass loss took place beneath the cosmic ray attenuation pathway, the recalculated average effective erosion rates are then 19.7 mm kyr$^{-1}$ and 8 mm kyr$^{-1}$ for the Pinedale and Bull Lake moraines, respectively. As there is no way to assess where this mass loss occurred, we instead utilize this degree of potential loss to place uncertainties on the average effective erosion rates in all

further calculations. These recalculations result in an average effective erosion rate decrease of 6% and 16% for the Pinedale and Bull Lake moraines, respectively.

## 3 Methods

### 3.1 $^{10}Be_{met}$ Analysis

We analyzed approximately 1-2 g aliquots of the <2 mm grain-size moraine sediment fraction from the same ~10-15 cm depth

intervals as Schaller et al. (2009a) analyzed for Be isotope abundance. We followed the sediment leaching procedure described in Ebert et al. (2012) and Wittmann et al. (2012), which was adapted from Bourlés (1988) and Guelke-Stelling and von Blanckenburg (2012), to extract Be isotopes from outer grain surfaces. Bulk samples underwent two steps to remove the adsorbed beryllium: a 24-hr agitation in 0.5 M HCl (to extract amorphous oxide-bound Be), and 1 M hydroxylamine-hydrochloride (to remove crystalline-bound Be). After each step, the supernate was separated from the sediment.


To measure the adsorbed $^{10}Be_{met}$, the leached material was homogenized with ~200 μl of $^9$Be carrier and 2 mL HF was added to the acid sample solution. This solution was nearly completely dried down and then dissolved in 1 additional mL of 50% HF acid and dried down completely. One additional mL of 50% HF acid was added and dried down completely again. We then added 10 mL ultrapure (18 MΩ) water to the warm fluoride residue and leached it for 1 h on a warm hotplate. The water

containing the Be was gently removed via pipette and dried down separately. The Be in the water leach solution was extracted and purified using a modification of the ion exchange chromatography procedure from von Blanckenburg et al. (2004) with minor adaptations for $^{10}Be_{met}$ purification. $^{10}Be_{met}/^9$Be ratios were measured at the Zurich AMS Lab (Kubik and Christl, 2010) (S555 standard, nominal $^{10}Be/Be = 95.5 \times 10^{-12}$), from which the $^{10}Be$ concentration ($^{10}Be_{reac} = {^{10}Be_{met}}$) was calculated. Two





carrier blanks analyzed with the samples register AMS $^{10}$Be/$^9$Be ratios of $3.2 \pm 1.5\times10^{-15}$, and $2.2 \pm 1.5\times10^{-15}$ containing

≪0.1% of the $^{10}$Be in analyzed samples.

## 3.2 $^{10}$Be$_{met}$ Flux Calculations

In an actively eroding setting, erosion rates can be calculated with knowledge of 1) the total inventory of $^{10}$Be$_{met}$ in the depth

profile, 2) a known/estimated $^{10}$Be$_{met}$ flux to the location, 3) the $^{10}$Be$_{met}$ retention behavior, and 4) an assumption of

approximate steady state conditions, which is only justified if the inventory of $^{10}$Be$_{met}$ is independent of the initial exposure

age of the soil. Here, steady state means that $^{10}$Be$_{met}$ lost through erosion and decay equals the $^{10}$Be$_{met}$ gained from atmospheric

flux (see *Willenbring and von Blanckenburg, 2010* for a full explanation and derivation), a prerequisite of which is that the

residence time of soil material containing meteoric $^{10}$Be with respect to erosion is much less than the depositional age

(Willenbring and von Blanckenburg, 2010). For an assumed steady state inventory, the inverse relationship between the local

erosion rate and the $^{10}$Be$_{met}$ content in the soil profile is exploited to determine a flux of $^{10}$Be$_{met}$ using the formulation of Brown

(1987), rearranged as follows:

$$F(^{10}Be_{met}) = E \times [^{10}Be]_{reac} + (I\lambda) \tag{1}$$

Where E is the erosion rate, $F_{(^{10}Be_{met})}$ is the atmospheric flux of $^{10}$Be$_{met}$ [atoms cm$^{-2}$ y$^{-1}$], I is the inventory of $^{10}$Be$_{met}$ [atoms cm$^{-2}$] in the depth profile, $\lambda$ is the decay constant of $^{10}$Be [y$^{-1}$], [$^{10}$Be]$_{reac}$ is the $^{10}$Be$_{met}$ concentration at the surface of the soil [atoms

g$^{-1}$], and $\rho$ is the regolith density [2.0 g cm$^{-3}$]. Inventories were calculated following Willenbring and von Blanckenburg (2010)

using a depth-averaged regolith density of 2.0 g cm$^{-3}$ for each profile (Schaller et al., 2009a,b), where *z* is the depth to the

bottom of the soil column and *[$^{10}$Be]$_{reac}$(z)* is the concentration of $^{10}$Be$_{met}$ at depth:

$$I = \int_0^z [^{10}Be]_{reac}(z)\rho dz \tag{2}$$

It is also possible to solve an equation for an erosion rate with only a $F_{(^{10}Be_{met})}$ estimate and a measured concentration of $^{10}$Be$_{met}$

at the soil surface. As with the inventory method above, this method also relies on the assumption that the system is at steady

state. Here, the effect of decay may also be ignored, as it becomes increasingly insignificant with increasing erosion rates, as

well as being minimal (1-4%) over the short lifespan of these moraines. Thus, $F_{(^{10}Be_{met})}$ may also be calculated by:



$$F(^{10}Be_{met}) = E \times [^{10}Be]_{reac} \times \rho \qquad\qquad (3)$$

Desorption of $^{10}Be_{met}$ can affect the inventory of $^{10}Be_{met}$ when erosion rates are low, water flux is high and soil chemistry favors mobility. Given that for these soil profiles pH ranges between 8 at depth and ~5.5 at the surface (Hall and Shroba, 1995), we must consider incomplete retention of beryllium and thus a reduced inventory used in (Eq. 1), and also reduced

surface concentration in (Eq. 3) (Bacon et al., 2012; Maher and von Blanckenburg, 2016). Applying a correction directly to the back-calculation of $^{10}Be_{met}$ flux is possible via (Eq. 3) of von Blanckenburg et al. (2012), which requires an accurate estimation of the water flux out of the system (Q) and the Be partition coefficient ($K_d$).

$$F(^{10}Be_{met}) = E \times [^{10}Be]_{reac} \times Q \times [^{10}Be]_{reac} \div K_d \qquad\qquad (4)$$


 $[^{10}Be]_{reac}$ is the equivalent of $N_{surf}$ in [Eq. 1, 3] and $K_d$ is estimated as $1 \times 10^5$ to $1 \times 10^6$ L/g from the surficial pH of ~5.5 via Be sorption-desorption experiments from You et al. (1989). We estimate Q by proxy via the modern precipitation rate of 0.276 (+ 0.055) m $y^{-1}$, with the uncertainty reflecting a modeled, time-integrated +20% paleo-precipitation rate for the Wind River Range from both 140ka- and 25ka- to present (Birkel et al., 2012) (See *Supplementary Material* for details).


Utilizing (Eq. 1), (Eq. 3), (Eq. 4) and *a priori* knowledge of the effective erosion rates, we back-calculate the loss-corrected $F(^{10}Be_{met})$ to the locations of these moraines after also accounting for inheritance (see *Sect. 5.1.1*). To further account for the full range of likely $K_d$, erosion rate (see *Sect. 2.2)*, and precipitation values, we employ Monte Carlo simulations and assess the likelihood and uncertainty of the best-fit fluxes.

**3.3 Calculated $^{10}Be_{met}$ Flux**

Monte Carlo simulations (n=100,000 iterations) across the range of all possible inputs to (Eq. 4) reveal a loss corrected $F(^{10}Be_{met})$ of 0.748 (+/-.069) x $10^6$ atoms $cm^{-2}$ $yr^{-1}$ and 0.779 (+.103 / - .09) x $10^6$ atoms $cm^{-2}$ $yr^{-1}$ for the Pinedale and Bull Lake moraines,



respectively. Each simulation yields a slightly positively skewed normal distribution (Fig. 3). The reported values represent the median $F_{(^{10}Be_{met})}$ from each simulation, with uncertainties representing their respective 95% confidence interval.


Retention calculations from (Eq. 4) indicate that the potential desorption loss at the surface of the Pinedale and Bull Lake profiles ranges from 0.7% to 6.5% and 1.7% to 14.7%, respectively.

These loss-corrected calculated fluxes are consistent for both moraines within uncertainty, using either base equation (Eq. 1

or Eq. 3) (differences between the two are virtually nonexistent, differing only by ~1%), which is consistent with the minor contribution by radioactive decay of $^{10}Be_{met}$ over the depositional time of the moraines. Our loss-corrected back calculations of $F_{(^{10}Be_{met})}$ from (Eq. 4), along with the range of possible independent flux estimates, are then normalized for paleomagnetic field intensity variations and compared in order to determine the best-fit flux to this area.

### 3.4 Independent $^{10}Be_{met}$ Flux Estimation

Accurately estimating $F_{(^{10}Be_{met})}$ from field experiments is a topic of ongoing debate (e.g. Ouimet et al., 2015; Dixon et al., 2018), particularly in regard to the effect of precipitation rate on the flux (i.e. whether precipitation leads to additive or dilution effects on delivered $^{10}Be_{met}$, see *Willenbring and von Blanckenburg (2010)* for a review). $F_{(^{10}Be_{met})}$ also varies through time, depending on solar and paleomagnetic field intensity, and has a spatial distribution primarily resulting from atmospheric mixing and scavenging. We thus compare our derived estimate of $F_{(^{10}Be_{met})}$ with other published estimates. One means to

estimate $^{10}Be_{met}$ production and delivery are $F_{(^{10}Be_{met})}$ estimates based on global atmospheric models (Field et al., 2006; Heikkilä and von Blanckenburg, 2015), which provide an estimate over large spatial scales. Another type of estimates is based on empirical, precipitation-dependent field estimates of $^{10}Be_{met}$ inventories in dated soils (Graly et al., 2011) measured over annual time scales. The work of Ouimet et al. (2015) highlighted the necessity for local $F_{(^{10}Be_{met})}$ estimates that also integrate over millennial time scales against models such as these, as their comparison of $^{10}Be_{met}$ inventories and deposition rates from



Pinedale- and Bull Lake-aged landforms in the Colorado Front Range exceeded those of the two approaches by an average of

30-50%.

The $F_{(^{10}Be_{met})}$ map of Heikkilä and von Blanckenburg (2015) utilizes the $^{10}Be_{met}$ production functions of Masarik and Beer

(1999) combined with the ECHAM5 general circulation model (GCM). Production rates were scaled to reflect the solar

modulation and magnetic field strength for the entire Holocene (280.94 MV) using measured $^{10}Be$ concentrations in ice cores.

The authors ultimately present a global grid of predicted "pre-industrial" and "industrial" (referring to simulated aerosol and

greenhouse gas concentrations) Holocene $F_{(^{10}Be_{met})}$ with an approximate cell size of 300 km x ~230 km. GCMs such as this are

useful for modelling atmospheric mixing of $^{10}Be_{met}$, particularly in the stratosphere, as well as the regional effect of climate

and its influence on $F_{(^{10}Be_{met})}$ via atmospheric circulation and precipitation (Heikkilä et al., 2012). At this latitude (~42.9° N),

the pre-industrial predicted $F_{(^{10}Be_{met})}$ of 1.38 x $10^6$ atoms cm$^{-2}$ yr$^{-1}$ is nearly identical to that derived from the flux map of Field

et al. (2006), which utilizes the GISS (Goddard Institute for Space Studies Model E) GCM to model production. While the

pre-industrial modeled $F_{(^{10}Be_{met})}$ is more applicable for comparison for landforms of these ages, we utilize the industrial

predicted $F_{(^{10}Be_{met})}$ of 2.37 x $10^6$ atoms cm$^{-2}$ yr$^{-1}$ as an upper bound uncertainty on their estimate.

On the other hand, the empirical, present-day estimates of $F_{(^{10}Be_{met})}$ from Graly et al. (2011) are based on measurements of

$^{10}Be_{met}$ deposition rates from contemporary measurements of $^{10}Be_{met}$ in precipitation, corrected for dust and normalized to a

modern (1951-2004) solar modulation value (700 MV). A first order estimate of the $F_{(^{10}Be_{met})}$ was empirically derived given

latitude (L) and average precipitation rate (P) to the study area (Graly et al., 2011):

$$F(^{10}Be_{met}) = P \times (1.44 / (1 + EXP((30.7 - L) / 4.36)) + 0.63) \qquad (5)$$



A predicted $F_{(^{10}Be_{met})}$ of 0.55 x $10^6$ atoms cm$^{-2}$ yr$^{-1}$ is calculated for these Wind River moraines using (Eq. 5), however in order

to compare these two estimates with each other, as well as to our calculated $F_{(^{10}Be_{met})}$, we normalized them all to a common

paleomagnetic intensity datum (i.e. the Holocene).

### 3.5 Normalizing flux estimates for geomagnetic intensity variations over the Holocene

Geomagnetic field strength has varied considerably from the late Pleistocene to present and exerts the primary quantifiable

influence on temporal variability in the production rate of cosmogenic nuclides in an inverse fashion (Pigati and Lifton, 2004).

Relative paleointensity over the last 140 kyr is, on average, ~20-40% of the current geomagnetic intensity depending on the

methodology employed (e.g. Frank et al., 1997; Valet et al., 2005). The flux map of Heikkilä and von Blanckenburg (2015)

accounts for paleomagnetic variations over the Holocene via the reconstruction method of Steinhilber et al. (2012), which

effectively increases the production rate used in their model by 1.23 times the present-day rate by rescaling the modern solar

modulation factor (Phi) and associated geomagnetic field intensity to that of the Holocene average (280.94 MV). As the

estimations of flux from Graly et al. (2011) were normalized to reflect a solar modulation of 700 MV, we rescaled the modern

Graly-derived $F_{(^{10}Be_{met})}$ to the average Holocene solar modulation factor of 280.94MV used in the flux map of Heikkilä and

von Blanckenburg (2015) via Masarik and Beer (2009) and Steinhilber et al. (2012). Thus, the predicted Holocene-average

$F_{(^{10}Be_{met})}$ using Graly et al. (2011) is 0.83 x $10^6$ atoms cm$^{-2}$ yr$^{-1}$ (Table 4). Similarly, we rescaled the calculated loss-corrected

$F_{(^{10}Be_{met})}$ for the Pinedale and Bull Lake moraines by integrating the production rate relative to the modern using the transport-

corrected $^{10}Be$ marine core record of Christl et al. (2010) from 25 ka and 140 ka, respectively, and propagating the statistical

uncertainties from the Monte Carlo simulations. Thus, the calculated best-fit Holocene-average $F_{(^{10}Be_{met})}$ are 0.92 (+ 0.083 / -

0.084) x $10^6$ atoms cm$^{-2}$ yr$^{-1}$ and 0.71 (+ 0.094 / - 0.084) x $10^6$ atoms cm$^{-2}$ yr$^{-1}$ for the Pinedale and Bull Lake moraines,

respectively (Table 4, Fig. 3).



# 4 Results

## 4.1 Cosmogenic Nuclide Concentrations

The measured $^{10}Be_{met}$ concentrations are reported along with the previously published $^{10}Be_{in\ situ}$ concentrations (Schaller et al.,

2009a) for the Pinedale and Bull Lake profiles (Table 1); $^{10}Be_{met}$ depth profiles are presented for the Pinedale and Bull Lake

profiles in Figure 2. The Pinedale depth profile has $^{10}Be_{met}$ concentrations ranging from $3.5 \pm 0.3$ to $200 \pm 6$ x $10^6$ atoms g$^{-1}$.

The highest nuclide concentration is measured at 10 cm, rather than at the surface. Below this maximum value, concentrations

decrease exponentially until reaching an asymptote at ~ 3 to 6 x $10^6$ atoms g$^{-1}$ from 43 cm to the bottom of the profile (180

cm), which we consider to be an inherited component (see *Sect. 5.1.1*). The Pinedale depth profile has an inventory of 6674 x

$10^6$ atoms cm$^{-2}$.

The Bull Lake depth profile has $^{10}Be_{met}$ concentrations ranging from $6.3 \pm 0.3$ to $415 \pm 12.5$ x $10^6$ atoms g$^{-1}$. The highest nuclide

concentration is measured at the surface; below this, concentrations decrease in an approximately exponential fashion until the

reaching an asymptote at ~ 6 to 8 x $10^6$ atoms g$^{-1}$ from 64 cm to the bottom of the profile (130 cm), which we also consider to

be an inherited component. The Bull Lake depth profile has an inventory of 19022 x $10^6$ atoms cm$^{-2}$. The $^{10}Be_{met}$ inventory

from the Bull Lake moraine is roughly 3 times higher than that of the Pinedale moraine.

All previous calculations derived from these $^{10}Be_{met}$ profiles have been corrected for their inherited component by subtracting

the lowest concentration of each profile from all depth interval concentrations, resulting in calculated flux rate differences of

less than 2%.

# 5 Discussion

## 5.1 Cosmogenic Nuclide Profiles

An approximately exponential decrease in $^{10}Be_{met}$ with depth is observed for the Pinedale and Bull Lake moraines (Fig. 2).

This trend can be explained most simply by the reactive transport of dissolved $^{10}Be_{met}$ with infiltrating water, as exponential

$^{10}Be_{met}$ profiles are predicted by reactive transport models (Maher and von Blanckenburg, 2016), however other possibilities

(e.g. remobilization of $^{10}$Be; diffusion of soil) exist (see *Willenbring and von Blanckenburg, 2010* for more detail).  We further

discuss the implications of these profiles in terms of degree of soil mixing in Sect. 5.1.2.

The maximum $^{10}$Be$_{met}$ concentration for the Pinedale moraine is measured at 10 cm depth, rather than the most surficial sample

(3 cm). This peak concentration corresponds with the clay rich layer of the B-horizon in the soil profile (Table 1). This

potentially indicates that this layer acts as a zone of eluviation, often observed in soil profiles that contain a mid-depth clay-

rich horizon (e.g. Monaghan et al., 1992). This subsurface maximum is usually a result of smaller grain sizes within this

horizon, as these grains have a higher surface area per unit mass and can exchange ions more easily (Brown et al., 1992;

Willenbring and von Blanckenburg, 2010). This phenomenon is not observed for the Bull Lake moraine; the highest clay

content observed in the profile is in the Bk-horizon at a depth of 43 cm (Schaller et al., 2009a,b), however no increase or

anomalous high $^{10}$Be$_{met}$ concentration is observed (Fig. 2, Table 1).

### 5.1.1 Inheritance

Analysis of the $^{10}$Be$_{in\ situ}$ depth profiles and calculated denudation rates of Schaller et al. (2009a) indicate that there is an

inherited nuclide concentration at depth, likely due to incomplete glacial erosion resetting for each moraine, that is higher for

the Bull Lake moraine ($\sim$ 1.2 - 1.8 x 10$^5$ atoms g$^{-1}$) compared to the Pinedale moraine ($\sim$ 0.3 – 0.6 x 10$^5$ atoms g$^{-1}$) (Fig. 2;

Table 1). The authors prescribe this observation to the presence of pre-irradiated reworked till in the Bull Lake moraine. The

existence of appreciable $^{10}$Be$_{met}$ concentrations at depth is also observed for the $^{10}$Be$_{met}$ depth profiles analyzed in this study,

with higher concentrations observed for the Bull Lake moraine (Fig. 2; Table 1), mimicking the trend seen in the $^{10}$Be$_{in\ situ}$

depth profiles. We also consider this incomplete resetting and presence of reworked till to be the predominant source of $^{10}$Be$_{met}$

at depth, as appreciable $^{10}$Be$_{met}$ mobilization to depth is unlikely for these profiles, as further explored below.

### 5.1.2 $^{10}$Be$_{met}$ Retention

A range of possibilities exist for retention effects and associated surficial $^{10}$Be$_{met}$ loss for these profiles. For the highest Kd

estimate, at 1 x 10^6 L/g, potential loss is as low as 0.7% and 1.7% for the Pinedale and Bull Lake profiles, respectively. On

the other hand, for the lowest Kd estimate, at 1 x 10^5 L/g, $^{10}$Be$_{met}$ loss due to desorption could be as great as 6.5% and 14.7%



at the surface of the Pinedale and Bull Lake profiles, respectively. Despite this, the Monte Carlo simulations indicate a low

probability for loss to this degree, particularly for the Pinedale profile, as evidenced by the relatively shallow tail for each

histogram (Fig. 3), with $F_{(^{10}Be_{met})}$ results within the 95% confidence interval. While the possibility of desorption cannot be

ruled out, we note that $^{10}Be_{met}$ mobilization to depth does not have an appreciable effect on $F_{(^{10}Be_{met})}$ in the vast majority of

simulations. Even in the worst-case scenario, the magnitude of the potential loss does not substantially affect our calculated

$F_{(^{10}Be_{met})}$ estimates within uncertainties.

### 5.1.3 Soil Mixing

The observed mixing depths for the Pinedale and Bull Lake terminal moraines as determined from the $^{10}Be_{in\ situ}$ depth profiles

of Schaller et al. (2009a) are between ~40 and 50 cm. The $^{10}Be_{met}$ concentrations gathered from the meteoric depth profiles

presented in this study do not show a similar homogeneity with depth near the surface. At first inspection, it appears that these

profiles do not exhibit the same mixing signal as those measured for $^{10}Be_{in\ situ}$, nor any mixing signal at all. In either case, either

the different grain sizes analyzed here and in Schaller et al. (2009a) exhibit different diffusion coefficients, or the advection of

$^{10}Be_{met}$ from the surface swamps the effect of mixing that is apparent in the $^{10}Be_{in\ situ}$ depth profiles. If the latter is the case, it

could indicate that continual reactive flow resets the $^{10}Be_{met}$ profile at timescales much shorter than that of physical mixing.

Profiles with a relatively low surficial pH (<5) might be particularly susceptible to this phenomenon due to incomplete retention

or differential mobility of $^{10}Be_{met}$ (Kaste and Baskaran, 2011), however the profiles analyzed here are not likely to show

appreciable (>15%) $^{10}Be_{met}$ loss due to retention issues.

### 5.2 $^{10}Be_{met}$ flux estimation; sources of variability

The best-fit, loss- and paleointensity-corrected $F_{(^{10}Be_{met})}$ of 0.92 (+ 0.083 / - 0.084) x $10^6$ atoms cm$^{-2}$ yr$^{-1}$ and 0.71 (+ 0.094 / -

0.084) x $10^6$ atoms cm$^{-2}$ yr$^{-1}$  for the Pinedale and Bull Lake moraines, respectively, are lower and higher, respectively,

compared to that estimated Graly et al. (2011), at 0.83 x $10^6$ atoms cm$^{-2}$ yr$^{-1}$, and are lower than that predicted by Heikkilä and

von Blanckenburg (2015), at 1.38 x $10^6$ atoms cm$^{-2}$ yr$^{-1}$ (Fig. 3; Table 4). The considerable discrepancy between the predicted

$F_{(^{10}Be_{met})}$ of each method arise primarily from differences in how each methodology treats the influence that precipitation rate



has on the flux to a given area and, in particular for this study, how large of an area is covered. The 310 km x 228 km flux map grid cell of Heikkilä and von Blanckenburg (2015) covers the entirety of the Wind River Range and the surrounding, relatively

low-lying flatlands (Fig. 1), where precipitation estimates vary considerably, by over an order of magnitude (WRCC, 2005), due to elevation and topographic effects on precipitation (Hostetler and Clark, 1997). For example, if one were to estimate $F_{(^{10}Be_{met})}$ from Graly et al. (2011) via (Eq. 5) to nearby Fish Lake Mountain contained within the same grid cell, with a modern precipitation rate of 128 cm $y^{-1}$, the $F_{(^{10}Be_{met})}$ would be 2.5 x $10^6$ atoms $cm^{-2}$ $y^{-1}$, substantially higher than that predicted from Heikkilä and von Blanckenburg (2015). Considering this alone, it is not surprising that such a discrepancy exists between

methods, nor is this a unique occurrence (e.g. Jungers et al., 2009; Schoonejans et al., 2017; Dixon et al., 2018).

Each approach has its own set of shortcomings, precluding agreement between each approach in sites such as this. The flux map of Heikkilä and von Blanckenburg (2015) has a coarse resolution and does not handle short wavelength orographic effects well, as observed in this comparison, along with being model based and requiring many assumptions on atmospheric

scavenging. The formula of Graly et al. (2011), on the other hand, does not take atmospheric circulation into account, instead relying on data from sites with relatively high rates of precipitation to derive an empirical formula. This approach may not prove to be wholly effective, as the magnitude of the ultimate influence of precipitation on $F_{(^{10}Be_{met})}$ is still unresolved (e.g. Willenbring and von Blanckenburg, 2010; Graly et al. 2011). A 100% additive precipitation-control on flux, if true, could lead to a higher predicted flux at sites with a higher time-integrated precipitation rate than historically recorded (up to 20%

higher in this case, see *Supplemental Material*; Fig. 3) from the method in Graly et al. (2011). Regardless, the strength of future $^{10}Be_{met}$ studies relies upon careful consideration of beryllium retention, spatial scale, and paleomagnetic intensity when determining $F_{(^{10}Be_{met})}$. As back-calculating a long-term delivery rate of $F_{(^{10}Be_{met})}$ for a particular site using $^{10}Be_{in\text{-}situ}$ and $^{10}Be_{met}$ is both costly and time-intensive, it is especially prudent to estimate $F_{(^{10}Be_{met})}$ using both methods compared here for robust $^{10}Be_{met}$ erosion rate calculations in the future.



## 6. Conclusions

In this study, we compare new meteoric $^{10}$Be and previously published *in situ*-produced $^{10}$Be depth profile measurements from the well-characterized Pinedale (~21-25 kyr) and Bull Lake (~140 kyr) moraines of Wind River, Wyoming. Our ability to utilize *a priori* knowledge of erosion rates from the $^{10}$Be$_{in situ}$ depth profile measurements of Schaller et al. (2009a), recalculated with revised parameters, allows us to back calculate best-fit, loss- and paleomagnetic intensity-corrected $^{10}$Be$_{met}$ fluxes of 0.92 (+ 0.083 / - 0.084) x $10^6$ atoms cm$^{-2}$ yr$^{-1}$ and 0.71 (+ 0.094 / - 0.084) x $10^6$ atoms cm$^{-2}$ yr$^{-1}$ to the Pinedale and Bull Lake moraines, respectively. Comparing this flux to two widely-used independent estimation methods reveals that the empirical flux estimate of Graly et al. (2011), after normalizing for Holocene paleomagnetic intensity, at 0.83 x $10^6$ atoms cm$^{-2}$ y$^{-1}$, is lower and higher for the Pinedale and Bull Lake profiles, respectively, and the modeled Holocene flux estimate of Heikkila and von Blanckenburg (2015), at 1.38 x $10^6$ atoms cm$^{-2}$ y$^{-1}$, is higher than the long-term calculated flux at this site. We find that loss of $^{10}$Be$_{met}$ in these profiles due to pH-influenced mobility/dissolution effects exerts a relatively minor potential control (biasing from 1%, up to 15%) on flux calculations. Inspection of the meteoric $^{10}$Be depth profiles and their near-surface concentrations suggest that soil mixing to depths of 40 and 50 cm, as observed for the Pinedale and Bull Lake $^{10}$Be$_{in situ}$ depth profiles, respectively, is not represented by the finer grain sizes analyzed in this study. The lack of a mixing signal may be most simply explained by different diffusion coefficients for the two grain size fractions, or from a swamping effect from the advection of $^{10}$Be$_{met}$ from the surface. These differences in the depth-concentration relationships between $^{10}$Be$_{met}$ and $^{10}$Be$_{in situ}$ might open up a new area of research to study particle movement in soils.

**Author Contribution**

TC is a current Ph.D. student at Scripps Institution of Oceanography and conducted the majority of the work during 2018-2019 under the supervision of JWK, who contributed to several drafts of the original manuscript as well as preparation of the meteoric data set. MS and JDB contributed via 10Be data acquisition, interpretation, and discussion; MC and PWK contributed via AMS measurements at ETH-Zurich. FvB assisted in interpretation of the comparative data set and associated discussion of meteoric 10Be flux estimates, mobility/retention, and paleomagnetic field intensity normalization.



**Competing Interests**

The authors declare no competing interests for this manuscript.

**Acknowledgement**

This work was made possible through the German Science Foundation Grant *BL562/7* and an Alexander von Humboldt Postdoctoral Fellowship. Support for TC came from a Career grant to Willenbring from NSF #1651243.

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





**Figures**

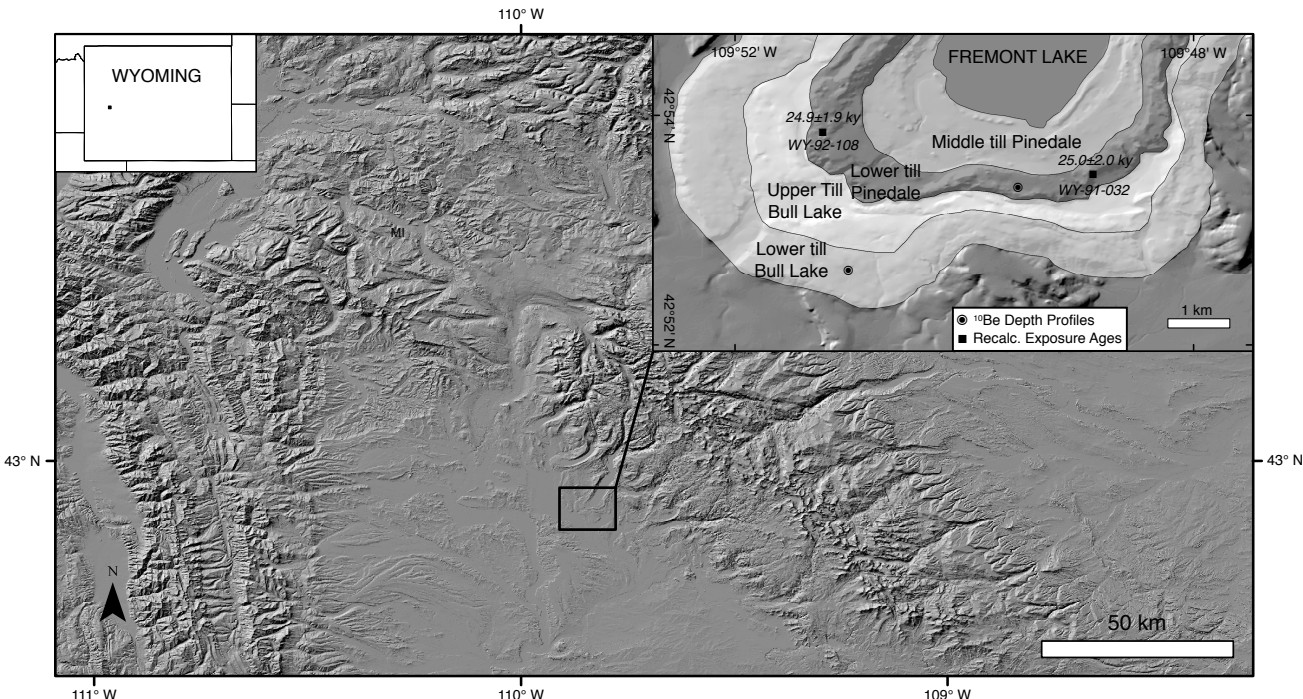

**Fig. 1)** Hillshade map of the Wind River range, derived from a 10 m digital elevation model (DEM); regional map encompasses the entirety of the meteoric [10]Be flux map grid cell of Heikkila and von Blanckenburg (2015). Inset (upper left) shows location
of regional map within Wyoming. Inset (upper right) shows locations of depth profiles analyzed for cosmogenic nuclide concentrations from the terminal Pinedale and Bull Lake moraines in the Fremont Lake area (after Richmond [1973] and Schaller et al. [2009a]). Also shown are the locations of boulder surface exposure dates for the Pinedale moraine (WY-92-108 and WY-91-032 of Gosse et al., 1995) that were recalculated using revised parameters (Table 2) to establish an updated independent age constraint for this moraine.






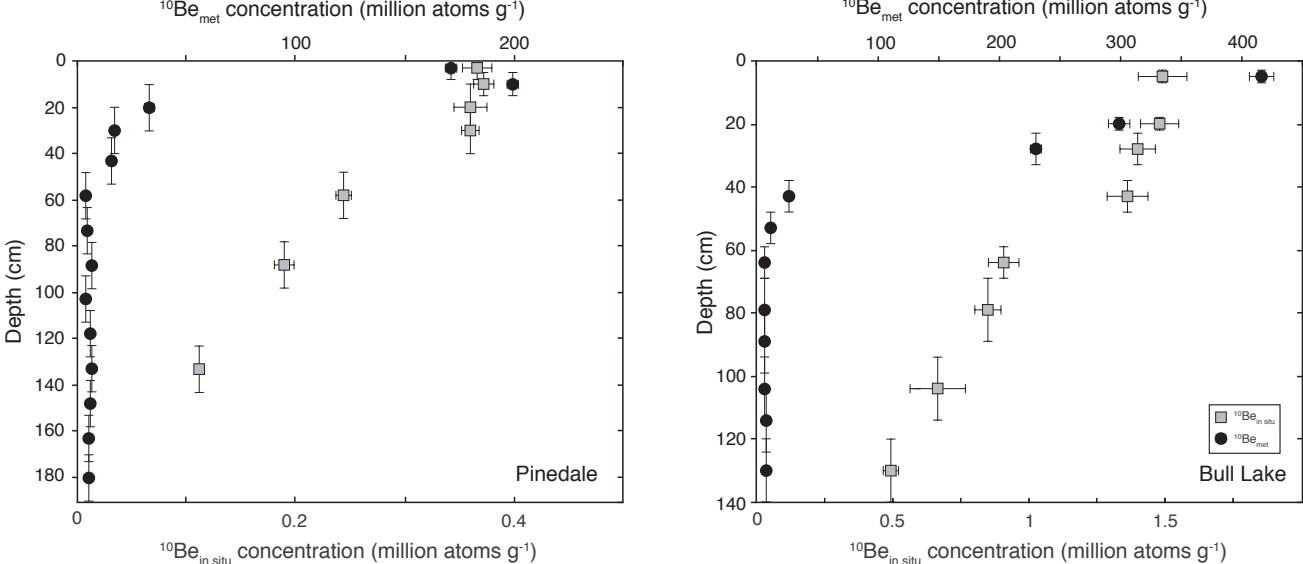

**Fig. 2)** (Left) Depth profile for the Pinedale moraine; $^{10}Be_{met}$ concentrations were measured from the <2 mm grain-size fraction of 14 samples from the same depth profile as analyzed for $^{10}Be_{in\ situ}$ in Schaller et al. (2009a). (Right) Depth profile for the Bull

Lake moraine; $^{0}Be_{met}$ concentrations were also measured from the <2 mm grain-size fraction of 11 samples from the same depth profile as analyzed for $^{10}Be_{in\ situ}$ in Schaller et al. (2009a). The $^{0}Be_{met}$ concentration at 94 cm was not measured.






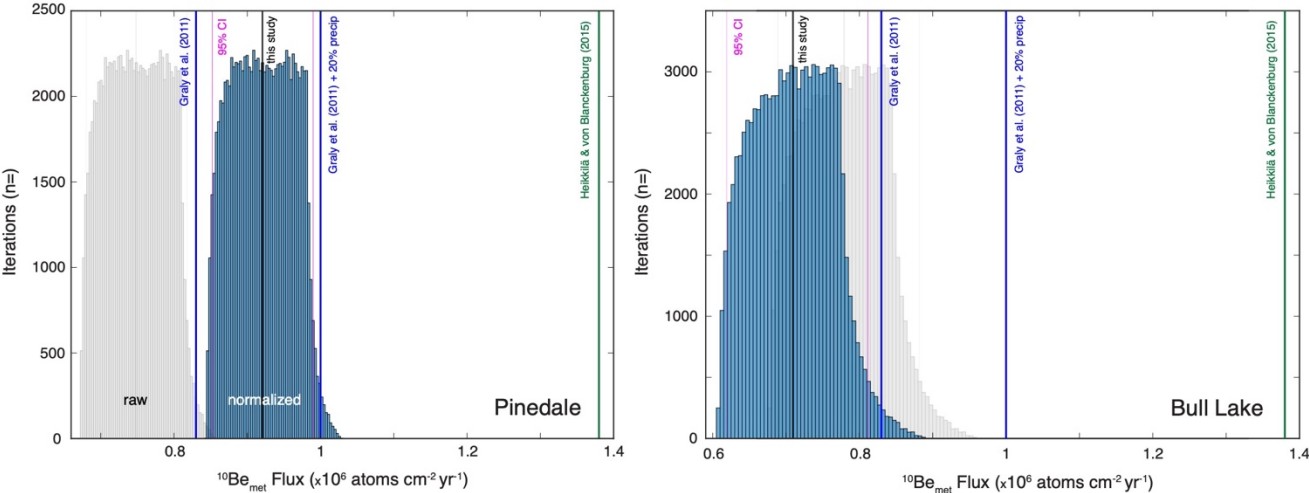

**Fig. 3)** Monte Carlo simulations (n=100,000) of all possible solutions for the flux of $^{10}Be_{met}$ ($F_{(^{10}Be_{met})}$) using (Eq. 4) for the Pinedale (left) and Bull Lake (right) profiles. Simulations before and after Holocene paleomagnetic normalizations are shown

alongside $F_{(^{10}Be_{met})}$ estimates from Graly et al. (2011) (with positive uncertainties representing the $F_{(^{10}Be_{met})}$ estimate with 20% greater precipitation rate, see text for details) and Heikkilä and von Blanckenburg (2015). Positive uncertainties for Heikkilä and von Blanckenburg (2015), representing their 'industrial' estimate, plot outside of each graph, at $2.37 \times 10^6$ atoms cm$^{-2}$ yr$^{-1}$. The best-fit $F_{(^{10}Be_{met})}$ for each profile is taken to be the median of all iterations, with uncertainties representing the respective 95% confidence interval for each simulation.


**Table 1.** $^{10}$Be Concentrations and GSD[a] in Depth Profiles from Pinedale and Bull Lake Moraines

| Sample[b] | Depth (cm) | Sand (wt %) | Silt (wt%) | Clay (wt %) | *In situ* $^{10}$Be concentration[c] ($10^5$ atoms g$^{-1}$) | Meteoric $^{10}$Be concentration[c] ($10^6$ atoms g$^{-1}$) | Meteoric $^{10}$Be inventory ($10^6$ atoms g$^{-1}$) |
|---|---|---|---|---|---|---|---|
| *Pinedale moraine (2262 m asl, 42° 53′ 26″ N, 109° 49′ 34″ W)* | | | | | | | |
| 04-WRMP-014 | 3 ± 2 | 75 | 18 | 6 | 3.67 ± 0.14 | 171.28 ± 5.14 | 1027.69 |
| 04-WRMP-013 | 10 ± 5 | 68 | 22 | 10 | 3.73 ± 0.09 | 199.53 ± 5.99 | 2793.37 |
| 04-WRMP-012 | 20 ± 10 | 70 | 23 | 7 | 3.6 ± 0.15 | 33.01 ± 3.18 | 660.15 |
| 04-WRMP-011 | 30 ± 10 | 74 | 22 | 4 | 3.6 ± 0.08 | 16.82 ± 1.54 | 336.39 |
| 04-WRMP-010 | 43 ± 10 | 76 | 19 | 5 | - | 15.36 ± 1.19 | 399.28 |
| 04-WRMP-009 | 58 ± 10 | 82 | 15 | 3 | 2.44 ± 0.07 | 3.97 ± 0.34 | 118.99 |
| 04-WRMP-008 | 73 ± 10 | 85 | 12 | 3 | - | 4.67 ± 0.38 | 140.18 |
| 04-WRMP-007 | 88 ± 10 | 81 | 16 | 3 | 1.89 ± 0.09 | 6.70 ± 0.56 | 200.96 |



| | | | | | | |
|---|---|---|---|---|---|---|
| 04-WRMP-006 | 103 ± 10 | 82 | 15 | 3 | - | 3.57 ± 0.32 | 107.06 |
| 04-WRMP-005 | 118 ± 10 | 71 | 23 | 6 | - | 6.21 ± 0.28 | 186.22 |
| 04-WRMP-004 | 133 ± 10 | 71 | 24 | 5 | 1.11 ± 0.03 | 6.49 ± 0.30 | 194.66 |
| 04-WRMP-003 | 148 ± 10 | 74 | 21 | 6 | - | 5.66 ±0.25 | 169.69 |
| 04-WRMP-002 | 163 ± 10 | 72 | 22 | 6 | - | 5.53 ± 0.24 | 165.94 |
| 04-WRMP-001 | 180 ± 10 | 72 | 23 | 6 | - | 5.10 ± 0.24 | 173.34 |
| | | | | | ∫ | 6673.92 |
| *Bull Lake moraine (2285 m asl, 42° 52' 39" N, 109° 51' 00" W)* | | | | | | |
| AT-FL-4L | 5 ± 2 | 69 | 22 | 9 | 14.9 ± 0.9 | 415.47 ± 12.46 | 4154.75 |
| AT-FL-4K | 20 ± 5 | 51 | 29 | 20 | 14.8 ± 0.7 | 298.81 ± 8.96 | 8964.39 |
| AT-FL-4J | 28 ± 5 | 52 | 34 | 14 | 14 ± 0.6 | 230.44 ± 6.91 | 3687.07 |
| AT-FL-4I[¤] | 43 ± 5 | 47 | 23 | 30 | 12.25 ± 0.7 | 26.59 ± 0.80 | 797.70 |
| AT-FL-4H | 53 ± 5 | 50 | 28 | 22 | - | 11.43 ± 0.34 | 228.67 |
| AT-FL-4G | 64 ± 5 | 54 | 26 | 20 | 9.08 ± 0.56 | 7.08 ± 0.38 | 155.82 |
| AT-FL-4F | 79 ± 10 | 60 | 24 | 16 | 8.5 ± 0.48 | 6.64 ± 0.23 | 199.19 |
| AT-FL-4E | 89 ± 10 | 62 | 24 | 14 | - | 6.32 ± 0.25 | 126.35 |
| AT-FL-4D | 94 ± 10 | 75 | 17 | 9 | - | 6.72[d] | 134.46[d] |
| AT-FL-4C[¤] | 104 ± 10 | 64 | 26 | 10 | 5.98 ± 1.0 | 7.13 ± 0.43 | 142.57 |
| AT-FL-4B | 114 ± 10 | 60 | 25 | 15 | - | 8.02 ± 0.24 | 160.42 |
| AT-FL-4A | 130 ± 10 | 60 | 25 | 15 | 4.93 ± 0.28 | 8.45 ± 0.25 | 270.37 |
| | | | | | ∫ | 19021.76 |

[a]Grain size distributions and *in situ* $^{10}$Be concentrations from *Schaller et al. (2009a)*

[b]See Schaller et al. (2009a) for the grain size fraction analyzed for each sample

[c]Corrected for blank, reported error includes analytical uncertainties (1σ)

[d]Average of $^{10}$Be$_{met}$ concentrations from directly above and below this depth

[¤]Average of multiple aliquots analyzed in Schaller et al. (2009a)





**Table 2.** Parameters for Recalculated *in situ* [10]Be Exposure Ages and Denudation Rates

| | General | Pinedale[1] | Bull Lake[1] |
|---|---|---|---|
| *Half Life (x $10^6$ yr)[2]* | 1.39 | | |
| *Decay Constant ($10^{-7}$ $yr^{-1}$)* | 4.998 | | |
| | | | |
| *Nucleonic production rate (at $g^{-1}$)[3]* | 3.92 | | |
| *Stopped muonic production rate (at $g^{-1}$)[3]* | 0.012 | | |
| *Fast muonic production rate (at $g^{-1}$)[3]* | 0.039 | | |
| | | | |
| *Nucleonic scaling factor* | | 6.631 | 6.7458 |
| *Stopped muonic scaling factor* | | 2.8093 | 2.8432 |
| *Fast muonic scaling factor* | | 2.8093 | 2.8432 |
| | | | |
| *Nucleonic adsorption length (g $cm^{-2}$)[4]* | 157 | | |
| *Stopped muonic adsorption length (g $cm^{-2}$)[4]* | 1500 | | |
| *Fast muonic adsorption length (g $cm^{-2}$)[4]* | 4320 | | |

[1]For denudation rate recalculations only; [10]Be exposure age recalculations are scaled
   by individual location via CRONUS-Earth web-based calculator (Phillips et al., 2016)

[2]from *Korschinek et al. (2010)*

[3]from *Borchers et al. (2016)* at sea level-high latitude

[4]from *Braucher et al. (2011)*







**Table 3.** Recalculated Best Fit Chi-Square Solutions for Different Denudation Rate Simulations of Schaller et al. (2009a)[1]

| Type of Denudation | Model | Age (kyr; *fixed parameter*) | Average Denudation (mm kyr⁻¹) | Inherited $^{10}$Be concentration ($10^5$ at g⁻¹) | Mixing Depth (cm) | Diffusivity $k$ ($10^{-3}$ m² a⁻¹) | Maximum Height (m) | Slope Angle (degrees) | Chi-Square Value |
|---|---|---|---|---|---|---|---|---|---|
| | | | *Pinedale moraine (2262 m asl, 42° 53' 26" N, 109° 49' 34" W)* | | | | | | |
| Constant | 2 | 25 | 15 | 0.2 | 0 | | | | 0.8 |
| Transient | 4 | 25 | 29-35 | 0.2 | 0 | 20 | 30 | 25,30 | 1-1.2 |
| | | | *Bull Lake moraine (2285 m asl, 42° 52' 39" N, 109° 51' 00" W)* | | | | | | |
| Constant | 6 | 140 | 7.5 | 1.4 | 0 | | | | 0.4 |
| Transient | 8 | 140 | 6-21 | 1.2-1.8 | 0 | 0.3-10 | 35,40,50,60 | 5,10,15, 20,25,30 | 0.3-0.4 |

[1]For a full explanation of range allowed in and resolution of each parameter, see Table 3 of *Schaller et al. (2009a)*

**Table 4.** Calculated $^{10}$Be$_{met}$ erosion rates for different flux estimates, raw and corrected for Holocene paleointensity variations

| Method | F($^{10}$Be$_{met}$) uncorrected (x $10^6$ atoms cm⁻²yr⁻¹) | Valid over time scale (kyr) | $^{10}$Be$_{met}$ correction factor relative to Modern | $^{10}$Be$_{met}$ correction factor relative to Holocene | F($^{10}$Be$_{met}$) corrected to represent Holocene (x $10^6$ atoms cm⁻²yr⁻¹) | Calculated Pinedale erosion rate (mm yr⁻¹) | Calculated Bull Lake erosion rate (mm yr⁻¹) |
|---|---|---|---|---|---|---|---|
| Pinedale (This Study) | 0.75 | 25 | 1.00" | 0.814" | 0.92 (+/- 0.08) | 26.9 (+/- 2.3) | - |
| Bull Lake (This Study) | 0.78 | 140 | 1.35" | 1.099" | 0.71 (+0.09 / - 0.08) | - | 8.5 (+1.1 / -0.9) |
| Graly et al. (2011)` | 0.55 (+0.11) | 0.005 | 0.82^ | 1.06* | 0.83 (+0.17) | 24.2 (+5) | 10 (+2) |
| Heikkilä and von Blanckenburg (2015)ᵗ | - | 10 | 1.23* | - | 1.38 (+0.99) | 40.3 (+28.9) | 16.6 (+11.9) |

\* *using the paleomagnetic reconstruction method of Steinhilber et al. (2012)*

^ *using the paleomagnetic scaling method of Masarik and Beer (2009)*

" *using measured $^{10}$Be$_{met}$ seafloor accumulation record of Christl et al. (2010)*

` *uncertainty represents +20% time integrated paleo-precipitation rate*

ᵗ *uncertainty represents the 'industrial' modeled flux of Heikkilä and von Blanckenburg (2015)*
