# Peer review of "Calibrating a long-term meteoric 10Be delivery rate into eroding Western US glacial deposits by comparing meteoric and in situproduced 10Be depth profiles"

_Geochronology, 2020_

## Referee Comment (RC1) · Anonymous Referee #1 · 27 May 2020

Having seen the presentation of this material at GSA 2018, I'm quite happy to see the authors have moved along to publication. The development of additional long-term records of 10Be retention in soil is one of the key pieces still needed for the robust application of the meteoric 10Be chronometer. This paper represents an important advance in that it factors erosion into the study. Previous work of this kind has relied on zero erosion assumptions that often complicate the interpretation of the results (e.g. Egli et al. 2010). I think the manuscript should be acceptable with minor to moderate changes. I detail my concerns by line or section below:

[Figure]

Line 25: Reword "Requires careful consideration" to something less vague.

Line 31: "Target atoms"?

Line 34: Also Al-OOH (e.g. Graly et al., 2010)

Line 43: I might avoid implying that most previous work is flawed here. These issues have been discussed and debated since the inception of the method with the work of Pavich and Monaghan.

Line 58 (and elsewhere): A priori knowledge refers to knowledge derived from first principles, etc. Data from another study is not a priori knowledge.

Line 60 (and elsewhere): Why "back-calculated", why not simply "calculated"?

Line 140: When you say "homogenized", do you mean that two aliquots from the sequential extraction were mixed together? I assume you must, since nowhere in the results do we see data from the separate sequences. This needs to be more clearly stated.

Line 156 (and elsewhere): I think "e.g. Willenbring and von Blanckenburg, 2010" would suffice. They were hardly the first or the only authors employ this concept of steady state (or the other concepts they receive sole credit for throughout the manuscript).

Line 173: You may ignore the decay effect, but must you?

Equations 3 and 4: I believe it is standard to use the interpunct for multiplication and a full line for division.

Equation 4 is wrong. The density term from Eq. 3 has disappeared, and the water flux term should be added not multiplied. I sincerely hope this is a typographical error, not an error that was implemented in the Monte Carlo model. But the authors should certainly double check this.

Line 192: The authors need to explain how they treated inheritance mathematically

(i.e. with an equation). The inherited fraction is also eroded and leached to depth, so it is not clear which approach was taken. I think inheritance should be included in equations 1-4, rather than tacked on separately without an equation.

Section 3.3: This section, as written, belongs in results not methods. In its place, a proper description of the Monte Carlo methods is needed. As it is, I don't see what the Monte Carlo accomplished that could not be done with error propagation.

Line 205: I am confused to as which equation (1,3, or 4) was actually used to generate the results presented. It sounds like all of them where, though the caption in figure 3 indicates eq. 4 was. The methods here need to be far more clearly presented.

Lines 210-220: This topic needs to properly treated in the introduction. The delve into the literature to characterize the "debate" and the various approaches is not appropriate to the methods section.

Section 3.5: The authors seem to take it as granted that paleomagnetic intensity exerts linear and predictable control on paleo 10Be flux. From what I can tell, this is far from certain. Looking at global datasets such as Frank et al. 2008, the two correlate but with significant deviation and scatter, including time periods (such as OIS 5e) where the correlation seems to break down entirely. I can't help but notice that the depositional fluxes derived from the two moraines are far closer to each other in raw form (Figure 3) than after paleomagnetic correction. What the authors seem to have done (line 259) is to simply use the average paleomagnetic intensity over the moraine age. But because erosion and leaching effects are cumulative, this should actually be weighted towards the more recent flux. If they wish to keep it all, the authors need to propagate the paleomagnetic flux correction through their model.

This section also seems to mix introductory background with methods and results.

Lines 269 & 276 / Table 1: The inventories should be reported at an appropriate precision and include propagated error calculations.

[Figure]

Line 278: I don't think the lowest concentration is the inheritance. The inheritance is the average of all of the values measured below the 60 cm (in this case).

Line 287 (and elsewhere): I personally find the need to call out other sections in advance to be a symptom of poor organization. The paper should flow naturally without the need to do this.

Line 293: Graly et al. 2010 tested this claim and found that grain size effects could explain subsurface maxima in none of the 29 soil profiles analyzed. A far better explanation is that 10Be is incorporated into the lattices of newly forming clays and oxyhydroxides at depth (e.g. Barg et al., 1997). Though in this case, the increase is fairly trivial and the depth and clay content small.

Section 5.1.2: This section would greatly benefit from having the Monte Carlo approach properly explained in the methods. As it is, the Monte Carlo is something of black box that gives surprising results on its own accord.

Line 319: Remove "At first inspection, it appears that".

Line 320: Remove "In either case".

Line 322: This is a surprising and novel observation that deserves further depth of treatment. Could you possibly mix coarse sand and fail to mix silt and clay? In some cases, patterned ground will mix pebble and cobble sized clasts at the hexagonal boundaries, excluding smaller grain sizes. Some delving into the cryoturbation literature seems warranted. Likewise, the second explanation needs further treatment. It is true that you only need to mix a declining profile for the in situ, whereas everything drops in at the top for the meteoric. But could you really homogenize one but not the other from these initial shape considerations alone? The reactive flow explanation proffered seems a bit wanting as well. How would reactive flow transport everything to the top of the otherwise mixing layer? This section would be much richer if a numerical model/calculation could be provided for any of these possibilities.

Line 348: An 100% additive precipitation control on flux is almost certainly not possible, as some dry deposition will occur, and complete scavenging and thereby dilution is likely in the largest storms. However, I think this is the wrong framework to consider. The paleo-precip factor is from a glaciological model and therefore quite uncertain. Nor is there any certainty in assuming that the "Graly Curve" for the Pleistocene was the same shape as that of the modern. Only after several more studies of this nature, will these sorts of things start to flesh out. I would recommend simply comparing to the modern and mentioning the paleo-precipation estimate in the discussion. But the second line on figure 3 and the "uncertainty" term on Table 4 seem to attribute too much to something we still know too little about.

Discussion: The deposition of recycled 10Be on dust is neglected in the analysis. Are there any estimates of Pleistocene dust flux in this region? If not, the uncertainty introduced by this unconstrained parameter should be at least mentioned.

The authors don't make any mention of the fact that their two moraines differ by a statistically significant margin. As I mention above, the difference is almost entirely due to the paleo-flux correction. So, if they keep the paleo-flux correction, they need to come up with something that varies in opposition to paleo flux to explain their results.

Line 636: "0Be"

Table 4: There is uncertainty inherent in the Graly curve apart from the + 20% attributed to paleo-precipitation. I believe this is true of the Heikkila GCM output as well. Per above, I think that simply treating the paleo-precipitation model as an upper bound is an overly credulous approach.

Supplement: I don't know why this information needs to be supplemental. The paper is not over long and I see no reason why this information cannot be integrated into the main text.

References Cited: Barg, E., D. Lal, M.J. Pavich, M.W. Caffee and J.R. Southon 1997.

Beryllium geochemistry in soils; evaluation of 10Be/ 9Be ratios in authigenic minerals as a basis for age models. Chemical Geology, 140: 237-258.

Egli, M., D. Brandová, R. Böhlert, F. Favilli and P.W. Kubik 2010. 10Be inventories in Alpine soils and their potential for dating land surfaces. Geomorphology, 119: 62-73.

Frank, M., B. Schwarz, S. Baumann, P.W. Kubik, M. Suter and A. Mangini 1997. A 200 kyr record of cosmogenic radionuclide production rate and geomagnetic field intensity from 10Be in globally stacked deep-sea sediments. Earth and Planetary Science Letters, 149: 121-129.

Graly, J.A., P.R. Bierman, L.J. Reusser and M.J. Pavich 2010. Meteoric 10Be in soil profiles – a global meta-analysis. Geochimica et Cosmochimica Acta, 74: 6814-6829.
* * *

---

## Short Comment (SC1) · 4 Jun 2020

The multiplication of the water flux term in Eq. 4 is a typographical error in the conversion from plain-text to 'Equation' in Word. It is meant to be an addition sign and has since been fixed. The Monte Carlo simulation uses Eq. 4 in the correct form.

Also, the density term should not be in Eq. 3. Density is factored into the erosion rate, which was not described properly with units (kg/m2/yr) on line 163 previously! This has been fixed as well.

---

## Referee Comment (RC2) · Anonymous Referee #2 · 30 Jun 2020

General comments

The goal of this study was to constrain the delivery rate/flux of meteoric 10-Beryllium (10Bem)to the Pinedale and Bull Lake glacial moraines at Fremont Lake in the Wind River range of Wyoming. The motivation was to improve the method of estimating the atmospheric 10Bem flux by implementing an erosion rate correction to previously established methods. The study area was selected because the deposit ages are known, and there is existing data on sediment grain size, weathering indices, soil properties, and erosion rates. The authors report results that both agree and disagree with pre-

vious flux estimates derived from other methods, which raises interesting questions about the controls on 10Bem in soils. This study is novel because it is the first to compare 10Bem and in situ 10Be for the same soil profiles. The proposed methods and results are important for the field of meteoric 10Be geochronology, and the results provide an opportunity to learn more about the behavior of 10Bem in soils and build on these new findings.

I believe the study merits publication after the authors have had the opportunity to make minor revisions. The manuscript text needs to be improved. The authors mix background information and results into the methods section, which makes it difficult to follow their approach. One of the primary equations used in this study, Equation (4), is incorrect as written, and it is unclear if the associated calculations were affected by this issue. The authors should re-check their calculations and edit this equation to make it dimensionally correct. The discussion section is fairly weak as is, and could be improved with sensitivity analyses that aim to identify the factors that most influence their results. The difference in results between the two moraines should also be further addressed (a sensitivity analysis could help with this comparison).

Specific and technical comments

General technical comment: There are numerous grammatical errors throughout the text. I recommend the authors read through the text carefully and fix places where there are missing words, or verbose text that could be made more concise.

Title: change "through a comparison of complimentary" to "by comparing complementary"

Line 23: How do these compare to the model fluxes of Heikkila and von Blanckenburg for the study area? Are they wildly different, or in close agreement? Would be good to mention this in the abstract for those readers who might use the modeled fluxes.

Line 24: Can the authors add the ages of these moraines to remind the reader over

what timescale they are averaging over for the fluxes?

Line 30: add uncertainty of +/-0.01 to (readers unfamiliar with 10Be might want to know the certainty of this half-life

Line 31: be more specific about which particles (i.e. 14N and 16O)

Line 34: Add both Al- and Fe-oxyhydroxides (Graly 2010 show that Al has a stronger relationship to 10Be concentrations)

Line 43: I would cite Graly et al 2010 who did an extensive analysis of the controls on 10Be concentrations in soil profiles from around the world.

Line 68: If it is windy, this implies either removal or deposition of fine particles over time, which could influence 10Bem concentrations. Can the authors say anything about dust delivery to this site?

Line 93 and 99: can the authors give uncertainty estimates, as this should factor into the uncertainty of their 10Bem delivery rates?

Line 108: change studies' to study's; and sites to site's

Lines 123-124: Why do the authors want to compare the Schaller denudations rates with 10Bem erosion rates? The 10Bem erosion rates (calculated using equations of von Blanckenburg et al., 2012) are not always comparable to denudation rates (they would need 9Be concentrations to calculate these rates). One could perhaps evaluate the chemical weathering component as the difference between the erosion and denudation rates.

Lines 128-130: Are there no major element data or weathering indices calculated for different depths within these profiles? In the introduction, the authors stated that they had all the data they needed to evaluate loss due to leaching and weathering.

Lines 138-139: Please mention that the amorphous and crystalline oxide fractions were re-combined before the next steps.

Line 141: ∼200 ul of 9Be carrier doesn't really provide any information because we don't know the concentration of the carrier solution. It's better to report the total mass of 9Be added to each sample.

Lines 142-143: Rather than repeating the previous sentence, say "The samples were then dried down and dissolved in an additional 1 mL 50% HF solution, repeated once."

Line 161: what unit do the authors use for erosion rate?

Line 165: rho is not used in equation (1), so the authors should introduce it in the next sentence, before equation (2). They also give the value for rho twice, but it is only needed once.

Equation (2): the authors should add in the correction for inherited 10Be into the equation.

Lines 172-173: It is best to include the decay effect in the equation. It might be negligible in this case, but may not be in older settings where this method may be applied in the future.

Line 181: use 'calculation' rather than 'back-calculation'

Equation (4): This equation is dimensionally incorrect as written. By rearranging Eq. 3 of von Blanckenburg et al. (2012), the erosion term should be added to the discharge term, not multiplied. It is also unclear what units the authors used for the variables because a water flux in m/yr does not cancel out with the partition coefficient, which is in L/g, unless a density term is inserted.

Line 186: The authors previously defined [10Be]reac, so they don't need to re-introduce it here. The authors also don't use the term 'Nsurf', which is from the Willenbring and von Blanckenburg (2010) equations.

Lines 160-194: The text would read more clearly if the authors first introduce the equations and variables, and then parameterize the equation in a paragraph following the

theory. If the authors change the format to theory first, followed application, it will be easier for the reader to follow the theory and then understand why and how each equation is applied.

Line 195: The calculated atmospheric 10Be flux estimates should be reported in the results section. It seems that the authors mix methods and results throughout the manuscript. These pieces should be separated.

Lines 210-233: This is all background information that should go in the introduction. The authors should place this information into context. What do we know about 10Bem atmospheric fluxes in the study area (e.g. from previous estimates, if existing, or from the GCM/GISS -based models)? The authors should identify the knowledge gaps highlighted by this background information, then pose their questions and hypotheses, and then go into the methods.

Lines 245-252: Similarly, the information about the variability in the geomagnetic field and its effect on 10Be atmospheric fluxes should e presented in the introduction, not the methods section. The authors should provide more detail on how the geomagnetic field strength influences the 10Be fluxes. Why is the modern solar modulation factor is much higher than the Holocene average? The authors should compare their Holocene-average flux of 0.92x106 at/cm2 yr to the value modeled by Heikkila and von Blanckenburg. If they are different, why? Could the dust flux make up an appreciable component of the Holocene-averaged flux? The authors should consider addressing this possibility in their flux reconstruction.

Line 249: The authors should mention which Heikkila and von Blanckenburg flux map (i.e. the pre-industrial map).

Line 280: The authors should include the inheritance correction in equations 1-4. Somewhere in the introduction, they should add that there is a high likelihood for inheritance since the concentrations were measured in reworked glacial till that may have been exposed to cosmic rays prior to burial.

Line 287: change parenthetical to: (e.g. Willenbring and von Blanckenburg, 2010)

Line 291: I believe the authors mean illuviation, rather than eluviation.

Lines 317-326: This paragraph raises a lot of questions about soil mixing, but leaves them mostly unresolved. Can the authors explore these questions in more detail? Because there is a low pH at the profile surface, can you estimate how much might be lost/mobilized down profile (e.g. based on Maher and von Blanckenburg, 2016 equations)? It appears that the grain size data in Tables 1 are from the <2 mm fraction only. How does the >2 mm size distribution change down profile? Could the relative abundances of pebble-sized clasts explain the difference between the in situ 10Be profile and the 10Bemet profile? It's possible that the fine fraction is relatively uniform down profile, but the coarse size fraction varies.

Lines 346-347: Can the authors provide some suggestions for resolving the influence of precipitation on F10Bemet? If this is identified as one of the key uncertainties influencing F10Bemet estimates, then they should provide a brief outlook for suture research into this topic.

Line 354: The authors do not make it exactly clear what two methods are being used to calculate the fluxes. Somewhere at the end of the introduction, the authors should state something along the lines of: "Here we estimate the atmospheric delivery flux of 10Bemet to the Wind River region using two methods: 1) ..., and 2) .... Then we compare the results of these methods to determine the best estimate for the local flux, and gain insight into the key processes regulating 10Be accumulation and retention in soil profiles so we can improve soil residence time studies."

Supplementary material: In the paleo-precipitation rates section, the reported 10Be flux values are missing the 'x106' term. Instead, they are reported as 1.09 and 0.66 atoms cm2 yr-1, respectively, which is impossibly low.

Figure 2: It would help to show corresponding plots of grain size data for these profiles

[Figure]

(e.g., wt% silt+clay). There is a typo after the semi-colon in the second sentence.

Table 2: If the methodology for the in situ exposure age and denudation rate calculations are in the supplement, then Table 2 should also go into the supplement.

Table 4: There are 10Bemet-derived erosion rates reported in this table, but neither the method nor the results are reported in the main text. The authors should add a section on the erosion rates and compare them to the in situ 10Be-derived erosion/denudation rates. This could make for an interesting comparison and ensuing discussion. The authors should also use numbers or letters for the superscripts in this table. Some of the chosen symbols could be confused with actual text.

References cited

Graly, J. A., Bierman, P. R., Reusser, L. J. & Pavich, M. J. Meteoric 10Be in soil profiles - A global meta-analysis. Geochim. Cosmochim. Acta 74, 6814–6829 (2010).

Maher, K. & von Blanckenburg, F. Surface ages and weathering rates from 10Be (meteoric) and 10Be/9Be: Insights from differential mass balance and reactive transport modeling. Chem. Geol. 446, 70–86 (2016).

Willenbring, J. K. & von Blanckenburg, F. Meteoric cosmogenic Beryllium-10 adsorbed to river sediment and soil: Applications for Earth-surface dynamics. Earth-Science Rev. 98, 105–122 (2010).

---

## Referee Comment (RC3) · Anonymous Referee #3 · 21 Jul 2020

This manuscript presents a careful attempt to use meteoric $^{10}Be$ ($^{10}Be_{met}$) concentration-depth profiles in deposits with known ages and denudation rates - Pinedale and Bull Lake moraines in Wyoming, USA - to calibrate long-term delivery rates (i.e., fluxes) of $^{10}Be_{met}$ to those sites. The authors leverage previous *in situ* $^{10}Be$ ($^{10}Be_{is}$) concentration-depth profiles and surface exposure ages (from boulders) to parameterize their calculations of $^{10}Be_{met}$ flux over tens to hundreds of millenia (the ages of the landforms being studied). This study carefully updates past dates and erosion rates according to advances in our understanding of both the half-life and production rate of $^{10}Be$, and it also diligently considers how factors such as precipitation rate and paleomagnetism may have varied over millennia affecting $^{10}Be_{met}$ flux, in turn. The authors then compare their site-specific results to two commonly used, empirically-derived methods for estimating $^{10}Be_{met}$ delivery rates, and they demonstrate that both methods overestimate flux rates for this site in Wyoming.

This work is one of only a couple studies that successfully constrain a delivery rate for $^{10}Be_{met}$, and (as the authors point out) knowing the delivery rate for a landscape of interest is of utmost importance if $^{10}Be_{met}$ is to be used as a tracer of surface process rates or as a geochronometer. This study represents an important contribution as a model for how long-term delivery rates for this isotopic system can be determined even when there is erosion of one's benchmark landform. That is to say that the authors show that it is not necessary to have all $^{10}Be_{met}$ retained since deposition in order to use a landform of known age as a calibration site. I think this article will be of interest to the readers of *Geochronology* from a methodological standpoint and also of interest to a broader audience interested in these iconic moraines of the Western United States.

I recommend this manuscript for publication after some minor to moderate revisions and clarifications. I lay out my thoughts/questions line-by-line below.

Line 34: Al-oxyhydroxides, too? See both Jungers et al., 2009 and Graly et al., 2011 in your references.

Lines 48-51: Consider rewording the sentence starting with "$^{10}Be_{met}$ shares a…" To me it is a little confusing and I think I only understand it because I'm already familiar with the differences between *in situ* and meteoric $^{10}Be$.

Line 68: I think you mean *a posteriori* here since the knowledge is based on empirical evidence. Could just simplify it to "...utilize previously determined effective…" Same spirit goes for other instances of *a priori* later in manuscript.

Line 68: "...50-year…" There are small grammatical and punctuation errors peppered throughout the manuscript. Nothing that derails the reading, but the authors should do a couple proofreads. I'll point out ones that jumped out. Not really being a grump here - just want to help.

Line 68: When talking about precipitation here, you are really reporting an annual *depth* rather than *rate* (as written).

Line 69: To me, the use of "proximal" here is confusing since that word has facies implications in geology. Just saying "nearby" might be clearer.

Line 78: Just to be clear, it sounds like you did not measure pH of your samples? I think it's reasonable to use the nearby measurements, although *in situ* pH measurements would be nice considering the potential impact on $^{10}Be_{met}$ mobility.

 Line 83: The suggestion here that the deepest samples are unweathered seems somewhat counter to the later argument that inherited meteoric concentrations are due to reworked material. Is there another model for inheritance that could work?

Line 92: I find "proximal" confusing again here, too (cf., Line 69). Do you mean nearby terraces or terraces that are proximal to the rangefront. Perhaps it doesn't matter, but I'd encourage precision with the language in both cases.

Line 95: The section that starts with "We recalculated…" seems like it should be part of the Methods section. There are several instances of methodology being presented either too early (such as here) or too late (such as the treatment of inherited concentrations), and I think that restructuring where these bits are presented would improve the clarity of the manuscript.

Line 102: Consider removing "...are likely…" All the moraines have experienced erosion since deposition.

Line 103: Stray hyphen in "...for-contiguous…"?

Lines 105-110: It seems like the averaging times of the methods may also play a role in the different results.

Line 113: "...were recalculated…" again suggests a section that may better fit in Methods. Some or all of the approach outlined in the Supplementary Materials could be integrated into the main text to good effect.

Line 116: I appreciate the consideration of transient denudation that you discuss here (in terms of a sensitivity analysis of your results), but you don't clearly justify why you set up the transient denudation the way you do. Why waning instead of cyclical, for example? Just justify your approach with a sentence and/or reference.

Line 130: "...erosion rate decrease…" From the original pub? Or is this the sum decrease of both recalculating and accounting for mass loss due to chemical weathering. Not immediately clear to me.

Line 147: "...minor adaptations…" Like what? You are so detailed in the preceding sentences, why not report your specific adaptations? Inquiring minds want to know!

Line 157: "...residence time...less than the depositional age…" I wonder if you can quantify this in some way to show that it holds for your site (seems like it certainly does). Can a residence time be inferred from the difference between your modeled flux rates and a "naive" flux rate determined by just dividing total inventory by moraine age. The discrepancy between those two numbers may be telling you something about how much $^{10}Be_{met}$ is being "lost" since deposition. Perhaps this isn't important, but it could be interesting in comparison to some of the diffusion modeling and other prior work that tried to quantify degradation rates for the moraines.

Line 163: Units for E?

Line 165: No *ro* term in Equation 1. I would recommend going through equations carefully to make sure they are correct. I imagine this is in the realm of typos rather than anything that made it into your modeling.

Line 175: Check unit analysis of Equation 3.

Line 186: There is no $N_{surf}$ in Equations 1 & 3.

Line 195: Section 3.3 reads more like Results rather than Methods.

Line 196: Nice agreement between flux rates! Remarkable stability over these timescales. Encouraging for future application of this isotopic system if one's local flux rate is known. Good stuff.

Line 204: I feel like I've lost track of what equations you are now reporting the results from. Perhaps a small table could clarify the differences between the outputs of Equations 1 vs. 2 vs. 4?

Line 216: "...type of estimate…" not "...type of estimates…"

Line 253: Should this bit about rescaling other approaches go into Methods?

Line 269: I think you really need to bring the discussion of potential inheritance into how you build your equations in your Methods. Can you just treat inheritance explicitly there? Then, in Results, you can certainly report apparent inheritance and discuss how that may occur.

Line 270: I believe there is a typo in your units for $^{10}Be$ inventory in Table 1. Check and correct.

Line 291: Think you mean "illuviation" not "eluviation" here. You are referring to removal of clay from above (eluviation) and the concentrating of clay in this horizon (illuviation).

Line 304: "...reworked till…" Just another flag to consider whether this idea of reworked till jives with the composition and state of weathering in your deepest samples.

Line 320: "...different diffusion coefficients…" Seems like this would manifest itself in some way beyond just the $^{10}Be_{met}$ depth profile. You'd see a trend in grain size with depth from the surface within the mixing layer or something. I think the difference between mixing timescales and the rate at which $^{10}Be_{met}$ is being translocated from the surface is more likely. For that matter, the formation of distinct clay horizons in at least the Pinedale suggests that soil horizonation happens faster than mixing (as inferred from the $^{10}Be_{is}$ profile). These are cool results with neat geomorphic and pedogenic process implications. Jungers et al., 2009, see a similar thing in hillslope soils of the Great Smoky Mountains.

Line 338: Where does the value of 128 cm/yr come from (in terms of both geography and a citation)?

Table 1: Check units for inventories in the final column.

Nice work - this is very cool stuff!

---

## Author Comment (AC1) · 22 Aug 2020

Thank you for your detailed review of our manuscript (gchron-2020-14) entitled Calibrating a long-term meteoric 10Be delivery rate into Western US glacial deposits through a comparison of complimentary meteoric and in situ-produced 10Be depth profiles.

The three reviewers provided great, thorough reviews which will enhance the readability and impact of this manuscript after revisions are made. We largely agree with the

majority of the reviewers comments and suggestions and summarize the final author 'major' comments for revisions as follows:

- The erosion rates used to calculate the meteoric fluxes are no longer the average between the constant and transient modeled denudation rates. Instead, we only use the average transient denudation rate (with uncertainties accounting for chemical weathering mass loss) for all calculations, as it is geologically incorrect to use the average rate between the constant and transient model runs – only one can be correct. We have added text to explain and justify this treatment, and have a note to the reviewers below that explains our rationale.

- Paleomagnetic intensity normalizations for the calculated fluxes for each moraine will now be calculated for the residence time of the soil profile down to the e-folding adsorption depth of meteoric 10Be (20 and 30 cm, and thus 6 and 24 kyr, for Pinedale and Bull Lake moraines, respectively) to properly weight and capture paleomagnetic variation effects on the production of meteoric 10Be over time (instead of over the entire ages of the moraines). The revised normalised meteoric fluxes now agree within uncertainty and are closer to the atmospheric model flux estimate. A table will be added to the Supplement that lists all factors employed in the Monte Carlo simulations, along with the MATLAB code used for the Monte Carlo simulations, so that future readers can also carry out calculations and normalize fluxes themselvess.

- The Monte Carlo approach will be properly introduced and described before presenting results. We will remove precipitation rate uncertainty (previously through an overly credulous paleo-precipitation rate estimation) in the simulation and associated text in the Supplement.

- All typographical errors will be fixed and reported units corrected for the main equations used for this work. Equation 4, which previously had a typo by which an addition sign was instead a multiplication sign, has been fixed. Equation 4 will also now include radioactive decay and meteoric inventory terms, and equation 3 will be removed.
This did not result in any appreciable change to our calculation results (as previously described).

- Soil mixing discussion will be combined with the section on Cosmogenic Nuclide Profile discussion and be expanded upon.

- The Introduction, Methods, and Results sections will be considerably re-organized so that there is no ambiguity between sections. This will enhance the readability and flow of this work substantially.

- We choose to leave our treatment of inheritance corrections as is, but will now explicitly define our treatment both qualitatively and analytically in the proper section.

See below for more detailed responses to your specific comments by line number. Please let us know if there are any questions about our suggested revisions.

Sincerely,

Travis Clow, Jane Willenbring, Mirjam Schaller, Joel Blum, Marcus Christl, Peter Kubik, and Friedhelm von Blanckenburg

Important note to reviewers and editor:

We have chosen to alter our approach regarding the known erosion rate for these moraines. Previously, we chose the known erosion rate as the average between the recalculated transient and constant denudation rate models of Schaller et al. (2009a) after accounting for potential chemical weathering mass loss. We have realized since our first submission that this is geologically incorrect – only one of the models can be valid – thus using the average between the two is erroneous. Instead, we now use the recalculated average transient denudation rates for all calculations, as this model is much more likely to be correct. Our justification is as follows:

Moraines are deposited in a triangular shape at the terminus of a glacier. Today they have more of a concave down parabolic shape. These two geometries have very different slopes and curvatures to them, which means the erosion rates must change through time. If you apply a linear (or nonlinear) hillslope diffusion law to understand moraine erosion, then the erosion rate equals the hillslope diffusivity of the moraine multiplied by the second spatial derivative of the topography (i.e. the curvature of the topography, or $dh/dt = k \, grad(h)$). Thus, the erosion rate depends on the curvature of the moraine topography.

Going back to the initial triangular shape of a moraine, the apex of the triangle (and the bottom corner where it sits on the ground) have the highest curvature when initially deposited. This part of the moraine will erode quickly at the start. As the apex flattens out and the bottom corners fill in, the curvature decreases, so the erosion rates will decrease. Erosion rates continue to decrease with time as a moraine flattens. Because of this, the erosion rate of moraines must be transient, with highest rates initially after deposition. All diffusion problems (e.g. temperature, hillslopes) respond this way (fast response at first, then slower response later) when adjusting to a non-equilibrium initial condition.

—

Response to Reviewer 1

Line 25: Reword "Requires careful consideration" to something less vague.

Removed 'careful'

Line 31: "Target atoms"?

Revised to "target nuclei"

Line 34: Also Al-OOH (e.g. Graly et al., 2010 )

Revised to "Fe- and Al-oxyhydroxides", citation added

Line 43: I might avoid implying that most previous work is flawed here. These issues have been discussed and debated since the inception of the method with the work of

Pavich and Monaghan.

Revised to "not all of which was possible in many of these studies"

Line 58 (and elsewhere): A priori knowledge refers to knowledge derived from first principles, etc. Data from another study is not a priori knowledge.

Revised to "previous knowledge" here and elsewhere.

Line 60 (and elsewhere): Why "back-calculated", why not simply "calculated"?

We initially chose to use the phrase 'back-calculated' as to be up-front that we are rearranging equations to solve for delivery rate (i.e. the calculated flux will always be a 'perfect match' for a known erosion rate), since all other meteoric studies to date utilize these equations to solve for erosion rate. However, this is a matter of taste, and can also be described as "calculated". We have since revised this to "calculated" here and elsewhere.

Line 140: When you say "homogenized", do you mean that two aliquots from the sequential extraction were mixed together? I assume you must, since nowhere in the results do we see data from the separate sequences. This needs to be more clearly stated.

Correct – the text now reflects this to be more clear.

Line 156 (and elsewhere): I think "e.g. Willenbring and von Blanckenburg, 2010" would suffice. They were hardly the first or the only authors employ this concept of steady state (or the other concepts they receive sole credit for throughout the manuscript).

Revised accordingly throughout the manuscript; added Brown et al., 1988 in this instance.

Line 173: You may ignore the decay effect, but must you?

No. Per Reviewer 2's comment on this matter, we now remove Eq.3 and include decay

and inventory in Eq. 4 for applicability to older settings.

Equations 3 and 4: I believe it is standard to use the interpunct for multiplication and a full line for division. Equation 4 is wrong. The density term from Eq. 3 has disappeared, and the water flux term should be added not multiplied. I sincerely hope this is a typographical error, not an error that was implemented in the Monte Carlo model. But the authors should certainly double check this.

The multiplication of the water flux term is a nasty typographical error. It has been fixed. Likewise, the density term should not be in Eq. 3. Density is factored into the erosion rate, which was not described with units (g/cm2/yr) on line 163 previously! This has been fixed as well.

Line 192: The authors need to explain how they treated inheritance (i.e. with an equation). The inherited fraction is also eroded and leached to depth, so it is not clear which approach was taken. I think inheritance should be included in equations 1-4, rather than tacked on separately without an equation.

Inheritance (lowest concentration measured) was subtracted from all concentrations measured. We have added text that defines this explicitly when defining [10Be]reac, as well as for the measured inventory, in this section.

Section 3.3: This section, as written, belongs in results not methods. In its place, a proper description of the Monte Carlo methods is needed. As it is, I don't see what the Monte Carlo accomplished that could not be done with error propagation.

This section has been moved to results – in its place is a proper description of the Monte Carlo simulation, which we use to determine uncertainties for our calculated fluxes. Traditional error propagation could also accomplish this goal. However, we do not have great constraints on Kd and evaluating the equation over the entire range of possible values in this manner provides a more realistic estimate on the uncertainties.

Line 205: I am confused to as which equation (1,3, or 4) was actually used to generate

the results presented. It sounds like all of them where, though the caption in figure 3 indicates eq. 4 was. The methods here need to be far more clearly presented.

In the interest of applicability of this method to both older and younger settings, we have removed Eq. 3 entirely and included inventory and decay in the old Eq. 4. We now explicitly state that this equation is used to generate the results presented.

Lines 210-220: This topic needs to properly treated in the introduction. The delve into the literature to characterize the "debate" and the various approaches is not appropriate to the methods section.

Agreed, majority of this section has been moved to the Introduction

Section 3.5: The authors seem to take it as granted that paleomagnetic intensity exerts linear and predictable control on paleo 10Be flux. From what I can tell, this is far from certain. Looking at global datasets such as Frank et al. 2008, the two correlate but with significant deviation and scatter, including time periods (such as OIS 5e) where the correlation seems to break down entirely. I can't help but notice that the depositional fluxes derived from the two moraines are far closer to each other in raw form (Figure 3) than after paleomagnetic correction. What the authors seem to have done (line 259) is to simply use the average paleomagnetic intensity over the moraine age. But because erosion and leaching effects are cumulative, this should actually be weighted towards the more recent flux. If they wish to keep it all, the authors need to propagate the paleomagnetic flux correction through their model. This section also seems to mix introductory background with methods and results.

The production rates dependence on paleomagnetic intensity is certainly not linear – even if calculated from paleomagnetic stacks using the "Elsässer formula". However we avoid doing this by instead reconstructing paleo-production from the measured 10Be stack (from marine cores) from Christl et. al (2010).

The comment that before correction, the depositional fluxes are quite close to one

another, yet deviate more after the correction however made us revisit the estimated time scale over which we have done this correction. This point is a great one that we agree with – that one should weigh these corrections towards the most recent flux. We now normalize over 6 ka and 24 ka for Pinedale and Bull Lake moraines, respectively, based on the residence time for the soil from the surface to the e-folding depth ($\sim$20 and $\sim$30 cm, respectively). This is a more realistic correction. Now, the corrected depositional fluxes for each moraine stay relatively close to each other (which one would expect for moraines so close to one another) and overlap within uncertainty.

After considering all reviewer comments and internal discussions, we have also decided to use the average transient erosion rate, recalculated from Schaller et al. (2009a), for all calculations, as it is more geologically correct than using the average between the constant and transient denudation rate model runs. Please see our note to the reviewers above with a detailed justification.

Lines 269 & 276 / Table 1: The inventories should be reported at an appropriate precision and include propagated error calculations.

This has been fixed.

Line 278: I don't think the lowest concentration is the inheritance. The inheritance is the average of all of the values measured below the 60 cm (in this case).

We chose to keep the lowest concentration as the inheritance to avoid negative inheritance-corrected 10Bemet measurements at depth.

Line 287 (and elsewhere): I personally find the need to call out other sections in advance to be a symptom of poor organization. The paper should flow naturally without the need to do this.

After re-organization of sections as advised by both reviewers, we have greatly reduced section callouts throughout the manuscript.

Line 293: Graly et al. 2010 tested this claim and found that grain size effects could

explain subsurface maxima in none of the 29 soil profiles analyzed. A far better explanation is that 10Be is incorporated into the lattices of newly forming clays and oxyhydroxides at depth (e.g. Barg et al., 1997). Though in this case, the increase is fairly trivial and the depth and clay content small.

This information has been added to the text as follows:

"This subsurface maximum could be the result of smaller grain sizes within this horizon, as these grains have a higher surface area per unit mass and can exchange ions more easily (Brown et al., 1992; Willenbring and von Blanckenburg, 2010). An alternative explanation invokes enhanced 10Bemet incorporation into the lattices of newly formed clays and oxyhydroxides at depth (e.g. Barg et al., 1997), though the increased clay content at this depth is not appreciably large."

Section 5.1.2: This section would greatly benefit from having the Monte Carlo approach properly explained in the methods. As it is, the Monte Carlo is something of black box that gives surprising results on its own accord.

We now include a paragraph introducing and summarizing the Monte Carlo simulation in the Methods section. We are also now explicit that the Monte Carlo simulation is used to determine the uncertainties on our calculated fluxes.

Line 319: Remove "At first inspection, it appears that".

Removed

Line 320: Remove "In either case".

Removed

Line 322: This is a surprising and novel observation that deserves further depth of treatment. Could you possibly mix coarse sand and fail to mix silt and clay? In some cases, patterned ground will mix pebble and cobble sized clasts at the hexagonal boundaries, excluding smaller grain sizes. Some delving into the cryoturbation literature seems warranted. Likewise, the second explanation needs further treatment. It is true that you only need to mix a declining profile for the in situ, whereas everything drops in at the top for the meteoric. But could you really homogenize one but not the other from these initial shape considerations alone? The reactive flow explanation proffered seems a bit wanting as well. How would reactive flow transport everything to the top of the otherwise mixing layer? This section would be much richer if a numerical model/calculation could be provided for any of these possibilities .

It is curious that smaller grain sizes wouldn't be mixed, but larger grains would – as finer grains are thought to have higher mobility than coarse grains and a tendency to migrate upward in a soil profile (e.g. Gray et al., 2020). A cryoturbation-related explanation would likely only explain a lack of mixing in the uppermost soil, however based on the in situ-produced 10Be data, we should expect a mixing signal down to $\sim$40 and $\sim$50cm for these profiles.

Reactive flow wouldn't transport everything to the top of the "in situ mixed layer", rather we are referring to a continual input of 10Bemet at the surface overwhelming any potential "meteoric mixed layer". In this case, even if there is mixing of these smaller grains going on in a similar fashion as the larger size fractions analyzed for in situ produced 10Be, before this mixed concentration can be set in stone, it gets 'reset' by the addition of newly-delivered 10Bemet , starting with the uppermost soil interval and then propagating to depth via reactive flow or possibly through macropore permeability. This requires reactive flow timescales to be much shorter than mixing timescales, which is a reasonable assumption given that typical soil mixing rates are low (cm's per century [Kaste et al., 2007]) versus the rapid adsorption of beryllium ($\sim$1 day [Boschi and Willenbring, 2016]) and fast permeability rates (as a rough proxy for reactive flow rates; $\sim$5 x 10^-3 m/s) for soils with these grain size distributions. We have revised the text to be more explicit about this. We are not certain how to numerically model such a scenario beyond back of the envelope style calculations by arbitrarily assigning diffusion and reactive flow rates (as described above), neither of which are known for this

site. Modeling using an existing framework like Be2D or LSD Mixing Model would be fantastic, but is not possible at present due to a higher degree of model sophistication needed, which is beyond the scope of this work.

Please note that we have now decided to merge the Soil Mixing and Cosmogenic Nuclide Profile discussion sections for consistency and readability.

Line 348: An 100% additive precipitation control on flux is almost certainly not possible, as some dry deposition will occur, and complete scavenging and thereby dilution is likely in the largest storms. However, I think this is the wrong framework to consider. The paleo-precip factor is from a glaciological model and therefore quite uncertain. Nor is there any certainty in assuming that the "Graly Curve" for the Pleistocene was the same shape as that of the modern. Only after several more studies of this nature, will these sorts of things start to flesh out. I would recommend simply comparing to the modern and mentioning the paleo-precipitation estimate in the discussion. But the second line on figure 3 and the "uncertainty" term on Table 4 seem to attribute too much to something we still know too little about.

Agreed, we have removed paleo-precipitation as an 'upper bound' in our calculations and instead mention the potential for paleo-precipitation rate to be higher in the text.

Discussion: The deposition of recycled 10Be on dust is neglected in the analysis. Are there any estimates of Pleistocene dust flux in this region? If not, the uncertainty introduced by this unconstrained parameter should be at least mentioned. The authors don't make any mention of the fact that their two moraines differ by a statistically significant margin. As I mention above, the difference is almost entirely due to the paleo-flux correction. So, if they keep the paleo-flux correction, they need to come up with something that varies in opposition to paleo flux to explain their results.

We mention on line 88 that eolian flux is insignificant, as determined by Sr isotope measurements of the moraine soils and dust sources from previous workers. Please see above for our new paleomag intensity normalization treatment.

Line 636: "0Be"

Fixed

Table 4: There is uncertainty inherent in the Graly curve apart from the + 20% attributed to paleo-precipitation. I believe this is true of the Heikkila GCM output as well. Per above, I think that simply treating the paleo-precipitation model as an upper bound is an overly credulous approach.

We have removed paleo-precipitation as an upper bound in the text, the table, and the MC simulation.

Supplement: I don't know why this information needs to be supplemental. The paper is not over long and I see no reason why this information cannot be integrated into the main text.

We have removed the section on paleo-precipitation rate, but have chosen to leave the Updated Independent Age Constraints section in the supplement as it is (necessarily) too detailed for the main text.

References Cited: Barg, E., D. Lal, M.J. Pavich, M.W. Caffee and J.R. Southon 1997. Beryllium geochemistry in soils; evaluation of 10Be/ 9Be ratios in authigenic minerals as a basis for age models. Chemical Geology, 140: 237-258.

Egli, M., D. Brandová, R. Böhlert, F. Favilli and P.W. Kubik 2010. 10Be inventories in Alpine soils and their potential for dating land surfaces. Geomorphology, 119: 62-73.

Frank, M., B. Schwarz, S. Baumann, P.W. Kubik, M. Suter and A. Mangini 1997. A 200 kyr record of cosmogenic radionuclide production rate and geomagnetic field intensity from 10Be in globally stacked deep-sea sediments. Earth and Planetary Science Letters, 149: 121-129.

Graly, J.A., P.R. Bierman, L.J. Reusser and M.J. Pavich 2010. Meteoric 10Be in soil profiles – a global meta-analysis. Geochimica et Cosmochimica Acta, 74: 6814-6829.

**GChronD**

---

## Author Comment (AC2) · 22 Aug 2020

Thank you for your detailed review of our manuscript (gchron-2020-14) entitled Calibrating a long-term meteoric 10Be delivery rate into Western US glacial deposits through a comparison of complimentary meteoric and in situ-produced 10Be depth profiles.

The three reviewers provided great, thorough reviews which will enhance the readability and impact of this manuscript after revisions are made. We largely agree with the

majority of the reviewers comments and suggestions and summarize the final author 'major' comments for revisions as follows:

- The erosion rates used to calculate the meteoric fluxes are no longer the average between the constant and transient modeled denudation rates. Instead, we only use the average transient denudation rate (with uncertainties accounting for chemical weathering mass loss) for all calculations, as it is geologically incorrect to use the average rate between the constant and transient model runs – only one can be correct. We have added text to explain and justify this treatment, and have a note to the reviewers below that explains our rationale.

- Paleomagnetic intensity normalizations for the calculated fluxes for each moraine will now be calculated for the residence time of the soil profile down to the e-folding adsorption depth of meteoric 10Be (20 and 30 cm, and thus 6 and 24 kyr, for Pinedale and Bull Lake moraines, respectively) to properly weight and capture paleomagnetic variation effects on the production of meteoric 10Be over time (instead of over the entire ages of the moraines). The revised normalised meteoric fluxes now agree within uncertainty and are closer to the atmospheric model flux estimate. A table will be added to the Supplement that lists all factors employed in the Monte Carlo simulations, along with the MATLAB code used for the Monte Carlo simulations, so that future readers can also carry out calculations and normalize fluxes themselvess.

- The Monte Carlo approach will be properly introduced and described before presenting results. We will remove precipitation rate uncertainty (previously through an overly credulous paleo-precipitation rate estimation) in the simulation and associated text in the Supplement.

- All typographical errors will be fixed and reported units corrected for the main equations used for this work. Equation 4, which previously had a typo by which an addition sign was instead a multiplication sign, has been fixed. Equation 4 will also now include radioactive decay and meteoric inventory terms, and equation 3 will be removed.
This did not result in any appreciable change to our calculation results (as previously described).

- Soil mixing discussion will be combined with the section on Cosmogenic Nuclide Profile discussion and be expanded upon.

- The Introduction, Methods, and Results sections will be considerably re-organized so that there is no ambiguity between sections. This will enhance the readability and flow of this work substantially.

- We choose to leave our treatment of inheritance corrections as is, but will now explicitly define our treatment both qualitatively and analytically in the proper section.

See below for more detailed responses to your specific comments by line number. Please let us know if there are any questions about our suggested revisions.

Sincerely,

Travis Clow, Jane Willenbring, Mirjam Schaller, Joel Blum, Marcus Christl, Peter Kubik, and Friedhelm von Blanckenburg

Important note to reviewers and editor:

We have chosen to alter our approach regarding the known erosion rate for these moraines. Previously, we chose the known erosion rate as the average between the recalculated transient and constant denudation rate models of Schaller et al. (2009a) after accounting for potential chemical weathering mass loss. We have realized since our first submission that this is geologically incorrect – only one of the models can be valid – thus using the average between the two is erroneous. Instead, we now use the recalculated average transient denudation rates for all calculations, as this model is much more likely to be correct. Our justification is as follows:

Moraines are deposited in a triangular shape at the terminus of a glacier. Today they have more of a concave down parabolic shape. These two geometries have very different slopes and curvatures to them, which means the erosion rates must change through time. If you apply a linear (or nonlinear) hillslope diffusion law to understand moraine erosion, then the erosion rate equals the hillslope diffusivity of the moraine multiplied by the second spatial derivative of the topography (i.e. the curvature of the topography, or dh/dt = k grad(h)). Thus, the erosion rate depends on the curvature of the moraine topography.

Going back to the initial triangular shape of a moraine, the apex of the triangle (and the bottom corner where it sits on the ground) have the highest curvature when initially deposited. This part of the moraine will erode quickly at the start. As the apex flattens out and the bottom corners fill in, the curvature decreases, so the erosion rates will decrease. Erosion rates continue to decrease with time as a moraine flattens. Because of this, the erosion rate of moraines must be transient, with highest rates initially after deposition. All diffusion problems (e.g. temperature, hillslopes) respond this way (fast response at first, then slower response later) when adjusting to a non-equilibrium initial condition.

—

Response to Reviewer 2

General technical comment: There are numerous grammatical errors throughout the text. I recommend the authors read through the text carefully and fix places where there are missing words, or verbose text that could be made more concise.

With the helpful suggestions and guidance from the reviewers, we believe all grammatical errors are now fixed in the revised manuscript.

Title: change "through a comparison of complimentary" to "by comparing complementary"

A welcome change! Revised.

Line 23: How do these compare to the model fluxes of Heikkila and von Blanckenburg

for the study area? Are they wildly different, or in close agreement? Would be good to mention this in the abstract for those readers who might use the modeled fluxes.

The calculated fluxes are both lower and higher than that estimated by Graly et al. (2011) for the Pinedale and Bull Lake moraines, respectively, and are lower than that predicted by Heikkila and von Blanckenburg (2015). This is a bit too specific for the abstract. Rather, we have revised the abstract text to note that a considerable discrepancy exists for both methods at this site, neither of which match the calculated fluxes within uncertainties.

Line 24: Can the authors add the ages of these moraines to remind the reader over what timescale they are averaging over for the fluxes?

Added

Line 30: add uncertainty of +/-0.01 to (readers unfamiliar with 10Be might want to know the certainty of this half-life

Added

Line 31: be more specific about which particles (i.e. 14N and 16O)

Added

Line 34: Add both Al- and Fe-oxyhydroxides (Graly 2010 show that Al has a stronger relationship to 10Be concentrations)

Added, along with citation.

Line 43: I would cite Graly et al 2010 who did an extensive analysis of the controls on 10Be concentrations in soil profiles from around the world.

Citation added.

Line 68: If it is windy, this implies either removal or deposition of fine particles over time, which could influence 10Bem concentrations. Can the authors say anything about dust

delivery to this site?

We note that dust delivery is insignificant to this site, based on Sr isotope measurements of these moraine soils and dust sources from previous workers, on line 88.

Line 93 and 99: can the authors give uncertainty estimates, as this should factor into the uncertainty of their 10Bem delivery rates?

The model of Schaller et al., 2009a does not permit for uncertainties in the independent age constraints when calculating denudation rates. These uncertainties only matter for the recalculated independent age constraints, and thus in situ produced 10Be effective erosion rates, which indirectly affect meteoric 10Be delivery rates. The flux we calculate solely depends on the estimate of denudation rate, not moraine age.

Line 108: change studies' to study's; and sites to site's

Fixed.

Lines 123-124: Why do the authors want to compare the Schaller denudations rates with 10Bem erosion rates? The 10Bem erosion rates (calculated using equations of von Blanckenburg et al., 2012) are not always comparable to denudation rates (they would need 9Be concentrations to calculate these rates). One could perhaps evaluate the chemical weathering component as the difference between the erosion and denudation rates.

We do not aim to compare the Schaller in situ denudation rates with meteoric erosion rates – we instead do as described – using the potential chemical weathering mass loss calculated by Schaller et al, 2009b to account for this component of the denudation rate of Schaller et al., 2009a in order to more properly compare "in situ-produced 10Be erosion rates" vs. meteoric 10Be erosion rates. Now that we use transient erosion rates for all calculations (see above), accounting for this potential chemical weathering mass loss is done so via the uncertainty for these transient erosion rates.

Lines 128-130: Are there no major element data or weathering indices calculated for

different depths within these profiles? In the introduction, the authors stated that they had all the data they needed to evaluate loss due to leaching and weathering.

Major element data is available from Schaller et al., 2009b, however we are unable to determine if this potential mass loss occurred above or below the cosmic ray attenuation pathway. The weathering rate is based on weathering loss in profile and material removed by denudation – with the rate based on the average of the four samples in the surface layer for each moraine (Schaller et al., 2009b). Since we do not know at which depths the material removed by denudation came from, we instead take this chemical weathering mass loss to be the uncertainty in the in situ-produced 10Be transient erosion rates used in all calculations.

Lines 138-139: Please mention that the amorphous and crystalline oxide fractions were re-combined before the next steps.

Added.

Line 141: âĹij200 ul of 9Be carrier doesn't really provide any information because we don't know the concentration of the carrier solution. It's better to report the total mass of 9Be added to each sample.

We now report the 9Be mass added.

Lines 142-143: Rather than repeating the previous sentence, say "The samples were then dried down and dissolved in an additional 1 mL 50% HF solution, repeated once."

Revised.

Line 161: what unit do the authors use for erosion rate?

g/cmˆ2/yr. This information has been added to the text, good catch.

Line 165: rho is not used in equation (1), so the authors should introduce it in the next sentence, before equation (2). They also give the value for rho twice, but it is only needed once.

This has been fixed.

Equation (2): the authors should add in the correction for inherited 10Be into the equation.

We instead now explicitly describe the inheritance correction before presenting these equations.

Lines 172-173: It is best to include the decay effect in the equation. It might be negligible in this case, but may not be in older settings where this method may be applied in the future.

Agreed, we have now removed Eq. 3. The old Eq. 4 has replaced Eq. 3 and has density and inventory terms accordingly.

Line 181: use 'calculation' rather than 'back-calculation'

We initially chose to use the phrase 'back-calculated' as to be up-front that we are rearranging equations to solve for delivery rate (i.e. the calculated flux will always be a 'perfect match' for a given erosion rate), since all other meteoric studies to date utilize these equations to solve for erosion rate. However, this is a matter of taste, and can also be described as "calculated". We have since revised this to "calculated" here and elsewhere.

Equation (4): This equation is dimensionally incorrect as written. By rearranging Eq. 3 of von Blanckenburg et al. (2012), the erosion term should be added to the discharge term, not multiplied. It is also unclear what units the authors used for the variables because a water flux in m/yr does not cancel out with the partition coefficient, which is in L/g, unless a density term is inserted.

The multiplication of the water flux term is a nasty typographical error. It has been fixed. Additionally, while we report discharge units as m/y, our calculations actually use L/m2/yr. Great catches – we have fixed these typographical issues and report units properly so everything is dimensionally consistent.

Line 186: The authors previously defined [10Be]reac, so they don't need to re-introduce it here. The authors also don't use the term 'Nsurf', which is from the Willenbring and von Blanckenburg (2010) equations.

This has been fixed.

Lines 160-194: The text would read more clearly if the authors first introduce the equations and variables, and then parameterize the equation in a paragraph following the theory. If the authors change the format to theory first, followed application, it will be easier for the reader to follow the theory and then understand why and how each equation is applied.

We chose to leave the format of this section as is. Aside from density in Eq. 2, we only directly parameterize Eq. 4, which already follows the theory at that point.

Line 195: The calculated atmospheric 10Be flux estimates should be reported in the results section. It seems that the authors mix methods and results throughout the manuscript. These pieces should be separated.

We have substantially re-organized the manuscript according to reviewer suggestions. Methods and results are now clearly separated – moving much of the background information (e.g. comment below) to the Introduction aided this process.

Lines 210-233: This is all background information that should go in the introduction. The authors should place this information into context. What do we know about 10Bem atmospheric fluxes in the study area (e.g. from previous estimates, if existing, or from the GCM/GISS -based models)? The authors should identify the knowledge gaps highlighted by this background information, then pose their questions and hypotheses, and then go into the methods.

The majority of this information has been moved to the Introduction and, in some instances, rephrased to reflect existing knowledge gaps (e.g. without a local calibration like we carry out in this paper, we do not have any way of knowing which production

rate estimation method is more correct – which is troubling for a site with such a discrepancy).

Lines 245-252: Similarly, the information about the variability in the geomagnetic field and its effect on 10Be atmospheric fluxes should e presented in the introduction, not the methods section. The authors should provide more detail on how the geomagnetic field strength influences the 10Be fluxes. Why is the modern solar modulation factor is much higher than the Holocene average? The authors should compare their Holocene-average flux of 0.92x106 at/cm2 yr to the value modeled by Heikkila and von Blanckenburg. If they are different, why? Could the dust flux make up an appreciable component of the Holocene-averaged flux? The authors should consider addressing this possibility in their flux reconstruction.

The average Holocene flux depends on variations in both solar modulation and magnetic field strength, which results in a flux that differs from modern.

Dust flux is insignificant at this site (as noted on line 88). We have moved the majority of this information to the Introduction, and instead present the estimated flux for this site in the Results section, and then speculate on differences between methods and the calculated flux in the Discussion.

Line 249: The authors should mention which Heikkila and von Blanckenburg flux map (i.e. the pre-industrial map).

We are using the pre-Industrial modeled flux, but we use the Industrial as an estimate of uncertainty. Text has been added to be more clear about this.

Line 280: The authors should include the inheritance correction in equations 1-4. Somewhere in the introduction, they should add that there is a high likelihood for inheritance since the concentrations were measured in reworked glacial till that may have been exposed to cosmic rays prior to burial.

We have added text that defines this explicitly when defining [10Be]reac, as well as for

the measured inventory. We have also added a sentence to the Introduction explaining the likelihood for inheritance in these deposits, as follows:

"We utilize bulk samples sieved to <2 mm for our analysis, extracted from the lower mineral soil developed on each moraine, both mixtures of reworked glacial till (composed of Archean granite, granodiorite, and dioritic gneiss) that have a high likelihood for inheritance from cosmic ray exposure prior to burial"

Line 287: change parenthetical to: (e.g. Willenbring and von Blanckenburg, 2010)

Fixed.

Line 291: I believe the authors mean illuviation, rather than eluviation.

Correct, this has been fixed.

Lines 317-326: This paragraph raises a lot of questions about soil mixing, but leaves them mostly unresolved. Can the authors explore these questions in more detail? Because there is a low pH at the profile surface, can you estimate how much might be lost/mobilized down profile (e.g. based on Maher and von Blanckenburg, 2016 equations)? It appears that the grain size data in Tables 1 are from the <2 mm fraction only. How does the >2 mm size distribution change down profile? Could the relative abundances of pebble-sized clasts explain the difference between the in situ $^{10}$Be profile and the $^{10}$Be$_{met}$ profile? It's possible that the fine fraction is relatively uniform down profile, but the coarse size fraction varies.

The equations of Maher and von Blanckenburg (2016) are for non-eroding settings. We can reasonably assume steady state for these profiles so using an upper and lower bound for $K_d$ is sufficient and achieves the same goal.

We do not have specific data on the GSD of the >2mm size distribution aside from it being assumed to be unweathered and representing $\sim$50% of the total material (Schaller et al., 2009b) – however, this wouldn't affect the in situ-produced $^{10}$Be profile any differently as those concentration measurements all came from the <2mm size fractions.

Lines 346-347: Can the authors provide some suggestions for resolving the influence of precipitation on F10Bemet? If this is identified as one of the key uncertainties influencing F10Bemet estimates, then they should provide a brief outlook for future research into this topic.

A hearty discussion on how to resolve this influence is beyond the scope of this work. However, new work by Deng and von Blanckenburg on this topic is about to appear in EPSL and we cite and summarize in a few sentences here.

Line 354: The authors do not make it exactly clear what two methods are being used to calculate the fluxes. Somewhere at the end of the introduction, the authors should state something along the lines of: "Here we estimate the atmospheric delivery flux of 10Bemet to the Wind River region using two methods: 1) . . ., and 2) . . .. Then we compare the results of these methods to determine the best estimate for the local flux, and gain insight into the key processes regulating 10Be accumulation and retention in soil profiles so we can improve soil residence time studies."

We have revised a couple of sentences to this effect at the end of the Introduction.

Supplementary material: In the paleo-precipitation rates section, the reported 10Be flux values are missing the 'x106' term. Instead, they are reported as 1.09 and 0.66 atoms cm2 yr-1, respectively, which is impossibly low.

Good catch! We have since removed this section, however.

Figure 2: It would help to show corresponding plots of grain size data for these profiles (e.g., wt% silt+clay). There is a typo after the semi-colon in the second sentence.

The inclusion of these plots tends to make this figure too busy. Instead, we report this data in Table 1, and if curious, the reader can compare to the GSD plots of Schaller et al. (2009a,b). Typo has been fixed.

Table 2: If the methodology for the in situ exposure age and denudation rate calculations are in the supplement, then Table 2 should also go into the supplement.

Table 2 has been moved to the Supplement.

Table 4: There are 10Bemet-derived erosion rates reported in this table, but neither the method nor the results are reported in the main text. The authors should add a section on the erosion rates and compare them to the in situ 10Be-derived erosion/denudation rates. This could make for an interesting comparison and ensuing discussion. The authors should also use numbers or letters for the superscripts in this table. Some of the chosen symbols could be confused with actual text.

We have decided to remove this section of Table 4, as we choose not to discuss them in the main text – erosion rates calculated from our study are circular, since we use them to calculate the depositional flux. They will always agree with the in situ derived rates. Any discussion therein is not warranted.
* * *

---

## Author Comment (AC3) · 22 Aug 2020

Thank you for your detailed review of our manuscript (gchron-2020-14) entitled Calibrating a long-term meteoric 10Be delivery rate into Western US glacial deposits through a comparison of complimentary meteoric and in situ-produced 10Be depth profiles.

The three reviewers provided great, thorough reviews which will enhance the readability and impact of this manuscript after revisions are made. We largely agree with the

majority of the reviewers comments and suggestions and summarize the final author 'major' comments for revisions as follows:

- The erosion rates used to calculate the meteoric fluxes are no longer the average between the constant and transient modeled denudation rates. Instead, we only use the average transient denudation rate (with uncertainties accounting for chemical weathering mass loss) for all calculations, as it is geologically incorrect to use the average rate between the constant and transient model runs – only one can be correct. We have added text to explain and justify this treatment, and have a note to the reviewers below that explains our rationale.

- Paleomagnetic intensity normalizations for the calculated fluxes for each moraine will now be calculated for the residence time of the soil profile down to the e-folding adsorption depth of meteoric 10Be (20 and 30 cm, and thus 6 and 24 kyr, for Pinedale and Bull Lake moraines, respectively) to properly weight and capture paleomagnetic variation effects on the production of meteoric 10Be over time (instead of over the entire ages of the moraines). The revised normalised meteoric fluxes now agree within uncertainty and are closer to the atmospheric model flux estimate. A table will be added to the Supplement that lists all factors employed in the Monte Carlo simulations, along with the MATLAB code used for the Monte Carlo simulations, so that future readers can also carry out calculations and normalize fluxes themselvess.

- The Monte Carlo approach will be properly introduced and described before presenting results. We will remove precipitation rate uncertainty (previously through an overly credulous paleo-precipitation rate estimation) in the simulation and associated text in the Supplement.

- All typographical errors will be fixed and reported units corrected for the main equations used for this work. Equation 4, which previously had a typo by which an addition sign was instead a multiplication sign, has been fixed. Equation 4 will also now include radioactive decay and meteoric inventory terms, and equation 3 will be removed.

This did not result in any appreciable change to our calculation results (as previously described).

- Soil mixing discussion will be combined with the section on Cosmogenic Nuclide Profile discussion and be expanded upon.

- The Introduction, Methods, and Results sections will be considerably re-organized so that there is no ambiguity between sections. This will enhance the readability and flow of this work substantially.

- We choose to leave our treatment of inheritance corrections as is, but will now explicitly define our treatment both qualitatively and analytically in the proper section.

See below for more detailed responses to your specific comments by line number. Please let us know if there are any questions about our suggested revisions.

Sincerely,

Travis Clow, Jane Willenbring, Mirjam Schaller, Joel Blum, Marcus Christl, Peter Kubik, and Friedhelm von Blanckenburg

Important note to reviewers and editor:

We have chosen to alter our approach regarding the known erosion rate for these moraines. Previously, we chose the known erosion rate as the average between the recalculated transient and constant denudation rate models of Schaller et al. (2009a) after accounting for potential chemical weathering mass loss. We have realized since our first submission that this is geologically incorrect – only one of the models can be valid – thus using the average between the two is erroneous. Instead, we now use the recalculated average transient denudation rates for all calculations, as this model is much more likely to be correct. Our justification is as follows:

Moraines are deposited in a triangular shape at the terminus of a glacier. Today they have more of a concave down parabolic shape. These two geometries have very different slopes and curvatures to them, which means the erosion rates must change through time. If you apply a linear (or nonlinear) hillslope diffusion law to understand moraine erosion, then the erosion rate equals the hillslope diffusivity of the moraine multiplied by the second spatial derivative of the topography (i.e. the curvature of the topography, or dh/dt = k grad(h)). Thus, the erosion rate depends on the curvature of the moraine topography.

Going back to the initial triangular shape of a moraine, the apex of the triangle (and the bottom corner where it sits on the ground) have the highest curvature when initially deposited. This part of the moraine will erode quickly at the start. As the apex flattens out and the bottom corners fill in, the curvature decreases, so the erosion rates will decrease. Erosion rates continue to decrease with time as a moraine flattens. Because of this, the erosion rate of moraines must be transient, with highest rates initially after deposition. All diffusion problems (e.g. temperature, hillslopes) respond this way (fast response at first, then slower response later) when adjusting to a non-equilibrium initial condition.

—

Response to Reviewer 3

Line 34: Al-oxyhydroxides, too? See both Jungers et al., 2009 and Graly et al., 2011 in your references.

Correct, this has been fixed; citations added.

Lines 48-51: Consider rewording the sentence starting with " 10 Be met shares a..." To me it is a little confusing and I think I only understand it because I'm already familiar with the differences between in situ and meteoric 10 Be.

This sentence has been reworded as follows:

10Bein situ shares a cosmic ray origin with 10Bemet but differs in production method; it is produced within crystal lattices in surface rocks and soil, rather than in the atmo-

[Figure]

sphere, with a well constrained total production rate of 4.01 atoms g-1 yr-1 at sea level, high latitude (Borchers et al., 2016), and is characterized by full retentivity and known production pathways with depth

Line 68: I think you mean a posteriori here since the knowledge is based on empirical evidence. Could just simplify it to "...utilize previously determined effective..." Same spirit goes for other instances of a priori later in manuscript.

Great catch – we have decided to change this to "previous knowledge" in all instances.

Line 68: "...50-year..." There are small grammatical and punctuation errors peppered throughout the manuscript. Nothing that derails the reading, but the authors should do a couple proofreads. I'll point out ones that jumped out. Not really being a grump here - just want to help.

Thank you – we have carefully re-examined and edited the text for these errors thanks to suggestions and catches like these from all reviewers. It is a bit embarrassing!

Line 68: When talking about precipitation here, you are really reporting an annual depth rather than rate (as written).

We have added explicit units of m a-1 here.

Line 69: To me, the use of "proximal" here is confusing since that word has facies implications in geology. Just saying "nearby" might be clearer.

Good call, this has been changed to "nearby" in all instances

Line 78: Just to be clear, it sounds like you did not measure pH of your samples? I think it's reasonable to use the nearby measurements, although in situ pH measurements would be nice considering the potential impact on 10 Be met mobility.

pH was unfortunately not measured in these samples :-(

Line 83: The suggestion here that the deepest samples are unweathered seems somewhat counter to the later argument that inherited meteoric concentrations are due to reworked material. Is there another model for inheritance that could work?

Not that we are aware of. That the deepest samples are unweathered is actually an assumption of Taylor and Blum, 1995 and is in reference to the >2mm size fraction, which is not what is analyze for either in situ-produced nor meteoric 10Be.

Line 92: I find "proximal" confusing again here, too (cf., Line 69). Do you mean nearby terraces or terraces that are proximal to the rangefront. Perhaps it doesn't matter, but I'd encourage precision with the language in both cases.

We have replaced proximal with nearby, as suggested.

Line 95: The section that starts with "We recalculated..." seems like it should be part of the Methods section. There are several instances of methodology being presented either too early (such as here) or too late (such as the treatment of inherited concentrations), and I think that restructuring where these bits are presented would improve the clarity of the manuscript.

Agreed, we have since restructured and re-organized this manuscript considerably based on all reviewer suggestions. This entire section is now in the Methods section.

Line 102: Consider removing "...are likely..." All the moraines have experienced erosion since Deposition.

Removed.

Line 103: Stray hyphen in "...for-contiguous..."?

Removed.

Lines 105-110: It seems like the averaging times of the methods may also play a role in the different results.

Indeed. We have added text to this effect.

Line 113: "...were recalculated..." again suggests a section that may better fit in Methods. Some or all of the approach outlined in the Supplementary Materials could be integrated into the main text to good effect.

This information has been reduced and moved to Methods – we chose to keep the Supplementary Material related to this there as it is (necessarily) overly detailed for the main text.

Line 116: I appreciate the consideration of transient denudation that you discuss here (in terms of a sensitivity analysis of your results), but you don't clearly justify why you set up the transient denudation the way you do. Why waning instead of cyclical, for example? Just justify your approach with a sentence and/or reference.

These scenarios are not prescribed by us, but rather by the model of Lal and Chen (2005) that Schaller et al. (2009a) uses. Nonetheless, we have added some additional text here describing the rationale they used in considering each scenario. As described in our note above, we have now chosen to use the average transient denudation rates (accounting for potential chemical weathering mass loss) for all calculations, instead of the average between the constant and transient denudation rates, as it is a more geologically sound approach. We now describe our rationale in the text.

Line 130: "...erosion rate decrease..." From the original pub? Or is this the sum decrease of both recalculating and accounting for mass loss due to chemical weathering. Not immediately clear to me.

We have removed this sentence from the manuscript.

Line 147: "...minor adaptations..." Like what? You are so detailed in the preceding sentences, why not report your specific adaptations? Inquiring minds want to know!

We have added this information to the text as follows:

The Be in the water leach solution was extracted and purified by a form of the ion exchange chromatography procedure from von Blanckenburg et al. (2004) that was

adapted for meteoric 10Be purification by passing the leachate through anion (2 ml of BioRad 1x8 100-200 mesh resin) and cation (2x 1 ml BioRad AG50-X8 200-400 mesh) exchange resins, precipitated at pH ~9 using NH4OH:H2O (1:1), washed twice with 2 ml ultrapure water with centrifugation in between, mixed with AgCl, centrifuged and dried overnight, and finally oxidized over open flame (>1000 °C; modified from Kohl & Nishiizumi, 1992).

Line 157: "...residence time...less than the depositional age..." I wonder if you can quantify this in some way to show that it holds for your site (seems like it certainly does). Can a residence time be inferred from the difference between your modeled flux rates and a "naive" flux rate determined by just dividing total inventory by moraine age. The discrepancy between those two numbers may be telling you something about how much 10 Be met is being "lost" since deposition. Perhaps this isn't important, but it could be interesting in comparison to some of the diffusion modeling and other prior work that tried to quantify degradation rates for the moraines.

We now calculate the residence time of the soil from the surface to the e-folding depth as 6 ka and 24 ka for the Pinedale and Bull Lake moraines, respectively, and use these timescales for paleomag normalizations.

Line 163: Units for E?

Added in (g/cm2/yr).

Line 165: No ro term in Equation 1. I would recommend going through equations carefully to make sure they are correct. I imagine this is in the realm of typos rather than anything that made it into your modeling.

It is factored into the erosion rate (which we neglected to define the units of) – this has been fixed.

Line 175: Check unit analysis of Equation 3.

Fixed.

Line 186: There is no N surf in Equations 1 & 3.

A remnant of an earlier draft of this manuscript – this has been fixed to 10Be[reac].

Line 195: Section 3.3 reads more like Results rather than Methods.

This section has been moved to Results. In its place is a proper explanation of the Monte Carlo simulation.

Line 196: Nice agreement between flux rates! Remarkable stability over these timescales. Encouraging for future application of this isotopic system if one's local flux rate is known. Good stuff.

It was quite a welcome surprise to us! Even after using the transient erosion rates instead of the average between the constant and transient erosion rates for our calculations, the raw flux rates still agree remarkably well.

Line 204: I feel like I've lost track of what equations you are now reporting the results from. Perhaps a small table could clarify the differences between the outputs of Equations 1 vs. 2 vs. 4?

We have removed Eq. 3, revised Eq. 4 to include decay and inventory, and are explicit that this equation is used for the MC simulation.

Line 216: "...type of estimate..." not "...type of estimates..."

Corrected.

Line 253: Should this bit about rescaling other approaches go into Methods?

Indeed – it has been moved to Methods.

Line 269: I think you really need to bring the discussion of potential inheritance into how you build your equations in your Methods. Can you just treat inheritance explicitly there? Then, in Results, you can certainly report apparent inheritance and discuss how that may occur.

Inheritance is now directly factored into the equations and reported accordingly in the Methods section.

Line 270: I believe there is a typo in your units for 10 Be inventory in Table 1. Check and correct.

Fixed.

Line 291: Think you mean "illuviation" not "eluviation" here. You are referring to removal of clay from above (eluviation) and the concentrating of clay in this horizon (illuviation).

Indeed, good catch!

Line 304: "...reworked till. . ." Just another flag to consider whether this idea of reworked till jives with the composition and state of weathering in your deepest samples.

See response to line 83 comment.

Line 320: "...different diffusion coefficients. . ." Seems like this would manifest itself in some way beyond just the 10 Be met depth profile. You'd see a trend in grain size with depth from the surface within the mixing layer or something. I think the difference between mixing timescales and the rate at which 10 Be met is being translocated from the surface is more likely. For that matter, the formation of distinct clay horizons in at least the Pinedale suggests that soil horizonation happens faster than mixing (as inferred from the 10 Be is profile). These are cool results with neat geomorphic and pedogenic process implications. Jungers et al., 2009, see a similar thing in hillslope soils of the Great Smoky Mountains.

Great inference – we agree that this is indeed a likely possibility and have added a couple sentences to this effect in the text. Please note that we have also decided to combine the Soil Mixing and Cosmogenic Nuclide Profile discussion sections for consistency and readability.

Line 338: Where does the value of 128 cm/yr come from (in terms of both geography

and a citation)?

Citation added

Table 1: Check units for inventories in the final column.

Fixed.

Nice work - this is very cool stuff!

Thank you!

---

## Author Response (AR3)

**GChron Response to Reviewers Round 2**

**Comments to the Author:**

**Dear Travis and colleagues,**

**Thank you for taking the time to substantially revise your manuscript. I asked two of the original referees to review your revised manuscript. You will see that one of these referees had additional concerns, particularly about eq. 4 and the combination of a transient erosion model with a steady-state flux calculation. I kindly ask you to consider these new reviewer comments in a revised version of the text. The other referee only had a handful of minor, technical corrections that should be easy to incorporate.**

**Thank you again for the hard work you have put into this manuscript thus far. Pending this second round of revisions, I believe your manuscript will be publishable in Geochronology.**

**All the best,**

**Marissa**

Hi Marissa,

Thank you again for facilitating this round of reviews. Below, we have addressed all of the concerns and comments from the reviewers, and feel we have properly justified:

1) Our updated approach of utilizing a transient erosion model with a steady-state flux calculation

2) That these moraines are indeed in steady state

3) That Eq. 4 is not in error

4) Our paleomagnetic normalization approach

To the last point, we have included a very detailed description of our approach along with a worked example -- both in the response to the reviewer, as well as in the Supplementary Material for the manuscript, including a helpful flowchart. Finally, we have decided to no longer use Monte Carlo simulations to determine uncertainties, as using traditional algebraic uncertainty propagation achieves the same result in a much simpler and easier to follow fashion. We believe our manuscript is now even stronger than before, and we thank the reviewers for their time and consideration in helping us improve this work.

Best,
Travis and co-authors

**Reviewer 1**

I have reviewed the revised text of "Calibrating a long-term meteoric 10Be delivery rate into eroding Western US glacial deposits by comparing meteoric and in situ produced 10Be depth profiles". The revised text is far clearer and better organized. However, as I now finally understand the methods employed, I have some new concerns to present.

First, the final term of equation 4 must be dropped. The 10Be that does not sorb in the topsoil layer would normally sorb at depth as infiltration continues and become part of the inventory. Under the steady state assumptions behind equation 4, 10Be delivered to the soil at depth is already counted in the loss to decay term. If the authors believe that 10Be is genuinely lost to ground water and not sorbed to the soil at any depth, the loss to ground water must be in terms of the entire inventory, not merely the surface layer. However, it is hard to imagine loss to groundwater at depth in this context, given the pedogenic carbonate build up in the soil. In principle, there could be a surface runoff term. But you'd need to calculate the proportion of precipitation that exits the system as overland flow, which I imagine is fairly small even in this semiarid context.

*The reviewer is right in pointing out that an assumption built into Eq. 4 is that all dissolved $^{10}$Be loss takes place at the surface. This is due to the steady state mixed reactor framework used by von Blanckenburg et al. (2012). This assumption becomes apparent as we use a surface $[^{10}Be]_{reac}$ to calculate a surface $[^{10}Be]_{diss}$ via Kd. The reviewer is also right to say that, if fluid is discharged from depth and $^{10}$Be is desorbed at depth, $[^{10}Be]_{diss}$ will be lower because $[^{10}Be]_{reac}$ is lower at depth (but it likely won't be zero, see below, nor do we see a large role for radiodecay given the young age of the moraines compared to the $^{10}$Be half life of 1.4My). Calculating this loss in terms of the inventory, as the reviewer suggests, would be best. However, this would require knowing the geometry and flux of fluid flow at any given depth, even to beneath the $^{10}$Be adsorption depth. This amounts to an impossible task.*

*The reviewer thus suggests removing the Q/Kd correction wholesale, thereby assuming that 100% of the delivered meteoric $^{10}$Be is adsorbed either at the surface or at depth. That assumption also does not hold true for this site, because of the pH (5.5 to 8 at depth) and associated Kd value here. Q/Kd is not sufficiently small compared to the erosion rate (von Blanckenburg et al,. 2012, Fig. A1). We thus cannot exclude the term and must consider retention for this to be a valid study.*

*In the revised version we thus emphasize that this loss correction represents an "maximum bound". Regardless, we note again that our retention calculations using Eq. 4 indicate that there is not substantial loss due to desorption. In the worst case scenario, with the lowest Kd estimate, the bias is only ~4% and ~9% for Pinedale and Bull Lake, respectively. At an average estimated Kd value (5.5E5 L/kg) for these sites, the bias is only ~1% and ~2%, respectively. We thank the reviewer for encouraging us to present a more explicit perspective of this previously implicit assumption.*

Secondly, I don't think it's reasonable to blithely combine a transient erosion model with a steady state flux calculation. The period of highest erosion would have occurred before

steady state was reached, while 10Be in the eroding topsoil was at significantly lower concentrations. The authors are therefore almost certainly overstating loss of 10Be to erosion and therefore overstating deposition. Furthermore, I doubt the 10Be profile in the Pinedale Moraine is anywhere close to steady state, even if erosion were steady (the Bull Lake may be). Depending on erosion rate and erosive depth, the time to steady state can potentially be hundreds of thousands of years (Graly et al., 2010). The authors need to create a transient model of 10Be development in the soil. I know they'd rather not, but there really isn't any way around this.

*This comment consists of two parts. Part 2 suggests that the meteoric $^{10}$Be is nowhere close to steady state and suggests that the time to steady state can be "hundreds of thousands of years". We strongly disagree with this assessment. We calculate the integration time scale of these erosion rates by dividing the adsorption depth (1/k = 20 and 30 cm for Pinedale and Bull Lake, respectively) by the erosion rate for each moraine and find that the integration time scales are 6 ky and 24 ky for Pinedale and Bull Lake, respectively. It's reasonable to assume that steady state is achieved after ~4-5 integration time scales have passed (Willenbring & von Blanckenburg, 2010), which corresponds to 24-30 ky for Pinedale, and 96-120 ky for Bull Lake. Given that the depositional age is 21-25 ky for Pinedale, and 140 ky for Bull Lake this justifies our consideration that these profiles are indeed in cosmogenic steady state. We have now made this point more apparent in the revised version.*

*Part 1 of the comment suggests that we should not calculate a meteoric $^{10}$Be flux (from Eq.4) using a constant erosion rate while at the same time basing this erosion rate on a transient erosion model. This is a valid statement to make. The reviewer also guesses correctly that we are indeed highly reluctant to design a transient model of meteoric $^{10}$Be accumulation, which would require a very substantial set of assumptions that would be close to impossible to constrain. A way around this would be to adopt the framework of Lal and Chen (2005), as Schaller et al. (2009a) did, but for meteoric $^{10}$Be, to constrain both age and erosion rate. However, their equations depend on the existence of a mixing zone, which we do not observe for meteoric $^{10}$Be in these profiles, so this is not possible. Thus, we maintain that while it is true that the transient erosion model indicates that erosion is fastest after initial deposition, the transient erosion rate modeled by Schaller et al. 2009, that we now use in this study, is not an "end-member rate" in the sense that the moraines were only eroding at 32 and 13 mm/ky for Pinedale and Bull Lake, respectively, during the initial wave of fast erosion, before the moraine evolved to the less flat-topped morphology we observe today. Instead, these rates are integrated over the entire age of the moraine, such that they capture the average of all erosion that has occurred at the moraine crests over these time periods. If we instead used the constant erosion model, we'd certainly be understating the loss of $^{10}$Be to erosion! Thus, we feel justified using the transient erosion model, as it gives us the most valid estimate of the true erosion rates for these landforms given what we know about hillslope diffusion. We have added explanatory text to section 3.2 making the reader aware of this potential complication as follows:*

> *"This approach integrates this transient behavior over the entire age of each moraines, and thus likely overstates the loss of $^{10}$Be to erosion to some degree, however it nonetheless provides the most realistic estimates possible for these moraines as we are otherwise unable to independently constrain their site-specific erosion rates"*

**Finally, the paleo-magnetic corrections remain poorly explained. No equations are provided nor is any data presented, save the final corrected numbers. A supplemental table that completely explains this is required.**

*All information that is needed to calculate the paleomagnetic corrections is in Table 3 (there was previously a typo for the correction factor relative to Holocene for Graly et al. 2011, however -- this was just a drafting error [not used in calculations] and has been fixed) and the references provided (Masarik & Beer 2009; Christl et al. 2010; Steinhilber et al. 2012). Maybe the reviewer is looking for a formula for converting paleomagnetic field strength and solar modulation into $^{10}$Be production. We do not use any. We simply linearly transform fluxes for a given integration time into another flux for another integration time, and to do so we use the graphs in Masarik & Beer (2009) and the conversion factors for the Holocene from Steinhilber et al. (2012), Fig. 3B. Please see our response to your line 287 comment below for a more detailed explanation. We added more explicit details of our treatment to the text as follows:*

> *As the estimations of flux from Graly et al. (2011) were normalized to reflect a solar modulation of 700 MV, we rescaled the modern Graly-derived $F_{(10Bemet)}$ to the average Holocene solar modulation factor of 280.94 MV used in the flux map of Heikkilä and von Blanckenburg (2015) following the paleomagnetic and solar intensity normalization procedure of Deng et al. (2020). This is carried out by first rescaling production at 700 MV to 500 MV (i.e. the modern solar modulation value of Steinhilber et al., 2012) via Fig. 4B of Masarik & Beer (2009) for a Graly et al. (2011)-specific modern scaling factor of 0.82 (Table 3). Then, to properly normalize for the Holocene, we multiply this modern scaling factor by the reciprocal of the rescaling factor of Heikkilä and von Blanckenburg (2015) (1.23) to arrive at a Holocene-normalized scaling factor of 0.67 and apply this to the Graly et al. (2011) flux estimate (Table 3). We illustrate and further describe the details of this procedure in the Supplementary Material (Fig. S1).*

*We have also added a flowchart to the Supplemental Material (Fig. S1) that takes the reader through each step of the calculation, with a worked example, to help illustrate and better describe the treatment. We feel this will be further beneficial to any reader that may want/need to carry out normalizations like these in the future.*

*To clarify how the moraine accumulation were corrected (integration time scales are 6 ky and 24 ky for Pinedale and Bull Lake, respectively) we have modified the text beginning line 251 as follows:*

> *"To further compare the model- and the precipitation-derived Holocene-average $F_{(10Bemet)}$ estimates with those calculated in this study, we must also normalize for geomagnetic and solar intensity variations within the Holocene (for Pinedale,with a 6 ky cosmogenic integration time) and beyond the Holocene (Bull Lake, with a 24 ky cosmogenic integration time). We again linearly rescaled our calculated loss-corrected $F_{(10Bemet)}$ for the and Bull Lake moraines by first integrating the production rate relative to the modern using the Principle Component 1 (PC1) of the $_{10}$Be marine core record of Christl et al. (2010), converting PC1 into relative fluxes from 6 ky and 24 ky, respectively, and then normalizing these values to those over the Holocene, propagating the statistical uncertainties. These time intervals represent the calculated residence/integration times of the soil profiles from the surface to the e-folding adsorption depth of $_{10}$Be$_{met}$ (20 and 30 cm for the Pinedale and Bull Lake moraines, respectively). This approach accounts for the*

*cumulative effects of transient erosion and leaching by weighting geomagnetic intensity variations on F(10Bemet) towards the present."*

**Some line-by-line comments:**

**116: No justification is given here for why the industrial run is a reasonable upper bound on the paleo 10Be fallout. Nor is it explained why industrial processes would make 10Be flux nearly a factor of 2 higher in this location.**

*Heikkila and von Blanckenburg (2015) explicitly describe how to determine uncertainty on their estimated fluxes -- it is the difference between the Modern and Pre-Industrial model runs, which is what we report as the uncertainty. Here is their explanation from their dataset:*

> *"Modern ("Industrial") Model: Direct output from ECHAM5-HAM modern atmosphere and aerosol loading was used from a 30 year run simulating the modern atmosphere characterised by industrial aerosol and greenhouse gas loading (Heikkilä et al., 2013a, Heikkilä and Smith, 2013c).*

> *"Pre-Industrial" model: The model is ECHAM5-HAM, run with preindustrial aerosol and greenhouse gas concentrations (Heikkilä et al., 2013b). The global flux of the pre-industrial model was adjusted to represent the same cosmic ray production rate as in the modern model.*

> *Average Model: The Modern and the Pre-Industrial model was combined by averaging. (Dark green Sheet). **The difference between both can be used as a rough uncertainty estimate** (light green sheet). The difference results from climate-dependent shifts in delivery of 10Be, but not on changes in its production."*

*We have added this information to the text as follows:*

> *"We use the pre-industrial modeled $F_{(10Bemet)}$ in our comparisons, as it is a more appropriate estimate for landforms of these ages. To place an upper bound uncertainty on this estimate, which is otherwise hard to quantify, we utilize the difference between the industrial and pre-industrial predicted $F_{(10Bemet)}$ (+0.99 x 10$^6$ atoms cm$^{-2}$ y$^{-1}$). This difference is solely a result of climate-dependent shifts in the delivery of $^{10}Be_{met}$ and shifts resulting from large industrial aerosol loading in modern times and does not reflect changes in atmospheric production (Heikkilä and von Blanckenburg, 2015)."*

**163-165: I find this statement deeply unsettling. Of course, you know where the mass loss occurred. That is the whole point of conservative tracer approaches. You know exactly, down to 10 cm scale, which elements leached out of the profile and in which abundances.**

*We are not entirely sure what point the reviewer is making here, nor what he means with "conservative tracer approaches", or how we would exactly know the locations of loss are. As the depth of loss is not well-constrained (i.e. with Tau depletion profiles) we cannot assume that all loss occurred beneath the in situ attenuation pathlength and instead apply the correction (calculated assuming loss completely beneath the in situ attenuation pathlength) to the uncertainties instead of directly to the average transient erosion rate. However, we assume that none of this affects the meteoric $^{10}$Be inventory (if this is what the reviewer means), for the*

*following reasons: Loss of meteoric $^{10}$Be does not need to depend on bulk weathering mass loss, but rather on surface sites available and pH. One extreme example: Assume you dissolve all plagioclase. You will have massive mass loss (Ca, Na, Si) but the clays that form and the Ca-buffered neutral pH ensure that all meteoric $^{10}$Be sticks by 100%.*

*We have changed this paragraph to the following text:*

> *"To properly compare the transient denudation rates of Schaller et al. (2009a) with the $_{10}Be_{met}$–derived erosion rates using the methods of von Blanckenburg et al. (2012), the weathering component of denudation must be accounted for. For the Pinedale moraine, chemical weathering mass loss is estimated to be 16% of the denudation rate, while for the Bull Lake moraine, the chemical weathering mass loss accounts for 20% (Schaller et al., 2009b). Assuming that the weathering mass loss took place beneath the cosmic ray attenuation pathway, the recalculated average effective transient erosion rates are then 27.0 mm ky$_{-1}$ and 9.9 mm ky$_{-1}$ for the Pinedale and Bull Lake moraines, respectively. As we have no means to assess whether this assumption is correct, we instead account for this degree of potential loss in the uncertainties (in addition to analytical uncertainties) on the effective transient erosion rates in all further calculations. Regardless, we note that such weathering mass loss does not necessarily need to coincide with loss of dissolved $^{10}Be_{met}$. Rather, the sites of primary mineral dissolution might also be the sites of secondary mineral formation and high dissolved Ca and hence potentially high $^{10}Be_{met}$ retentivity."*

**174: This still doesn't clearly state that the two aliquots were combined. Maybe "combined and homogenized".**

*Revised to "combined and homogenized".*

**Equation 3: In the previous round, I asked the authors to explicitly add inheritance to this equation. I don't know why they haven't. Explaining something in the text but omitting to include it in formal terms is not enough.**

*Inheritance corrections are now directly included in Eqs. 2, 3, and 4*

**Equation 4: The first two terms do not need to be in parentheses. Neither does Iλ (also in eq. 2).**

*Parentheses removed.*

**235: So instead of Gaussian, you assume the value is equally likely to fall anywhere within the confidence interval? This doesn't seem justified. You should model the actual probability distribution of each of your values and randomly select from these (if you don't want to solve it analytically).**

*This is indeed what we previously did, although we have decided to alter our approach as originally suggested by the reviewer in the last round (see below).*

*For our previous treatment -- yes, Kd values are equally as likely to fall anywhere within the range of values described, with the previous knowledge we have available. For justification -- see below for a plot of Kd values estimated from the Be sorption-desorption experiments from You et al. (1989). The soil profiles analyzed here have an estimated pH of 5.5 at the surface to 8 at depth. Any of these estimates are possible.*

[Figure]

*Despite this, it's important to note that uncertainties determined via traditional error propagation, assuming a Gaussian distribution for Kd (and transient erosion rates), for the Pinedale and Bull Lake loss-corrected fluxes are essentially the same as before, now at 1.04 x 10$^6$ +/- 0.14 at/cm$^2$/y and 1.04 x 10$^6$ +/- 0.39 at/cm$^2$/y, respectively. Please note that these raw (i.e. paleomag-uncorrected) values differ very slightly from the last submission because of the Gaussian treatment. That the estimates are now equal is novel, but unsurprising, given that they previously also agreed very well within error.*

*We have thus decided to do away with the Monte Carlo entirely, as it makes the determination of uncertainties more complicated than it needs to be to arrive at essentially the same answer. Previously, because we analytically solved (Eq. 4) using an average value for Kd (instead of the median result of the Monte Carlo), we ended up with lopsided uncertainties which is admittedly*

*confusing (see response below). Since there is no appreciable change in uncertainties with error propagation (nor should there be), the simplest route is the best -- as the reviewer indeed insinuated with the first round of reviews. We have removed all text regarding Monte Carlo-derived uncertainties and have replaced (where necessary) with text describing our error propagation approach.*

**243: Do you mean 20-40% less than current?**

*Good catch! This has been corrected.*

**278: I don't understand why the uncertainties are so lopsided (especially for Pinedale), when the Monte Carlo results are so flat.**

*This is due to how we use the Monte Carlo -- to only solve for uncertainties, not the reported flux value itself. The median result of the Monte Carlo is not the same result as solving the equation with average values for Kd and erosion rate, which we previously reported.*

*However, given that determining uncertainties via traditional error propagation gives virtually the same result as the Monte Carlo-derived uncertainties, we have chosen to remove the Monte Carlo from the manuscript and instead report uncertainties using traditional algebraic error propagation.*

**282: As I mention above, these calculations are in error and need to be excised.**

*These calculations are not in error - they represent potential maximum bounds for loss. Please see our above justification.*

**287: I am totally mystified as to why this correction is so large. The records presented in Christl et al., 2010 show very little in the way of variation over the past 25 ka. We need to see the data and equations behind this.**

*The correction is so large because we must scale for the Holocene after scaling relative to Modern. As shown in Table 3, the scaling factor relative to the modern (determined using Christl et al. 2010, as described in the table) is not that large -- in fact, it's only 1% less than modern for Bull Lake (which is integrated over 24 ky). This is exactly as you describe -- little variation over the past 24 ky.*

*For the Graly estimate, we cannot use Christl et al. 2010 to normalize to Modern. Instead, we must first recalculate the solar modulation they used (700MV) relative to a phi of 500 MV (this is modern solar modulation value of Steinhilber 2012, which is also used for H&FvB's rescaling in their flux map, by which they then rescale to 280 MV to represent the Holocene average). We do so by utilizing Fig. 4b of Masarik & Beer (2009). The production at 700 MV at this latitude is*

*0.018 atoms/cm$^2$/s$^1$, at 500 MV at this latitude it is 0.022 atoms/cm$^2$/s$^1$, dividing the former by the latter gives the production ratio relative to modern of 0.82 for the Graly estimate.*

*However, we cannot stop here. We must then scale this value to the Holocene so that \*all\* flux estimates (including our calculations) agree, otherwise we cannot properly compare between the three flux estimates. This is done by multiplying the scaling factor relative to Modern for our calculations (0.88, 0.99 for Pinedale and Bull Lake) as well as Graly et al. (2011)'s (0.82) by the reciprocal of the scaling factor used by Heikkila and von Blanckenburg (2015) (1.23). This effectively scales all estimates to the average Holocene solar modulation factor of 280.94, as is already done with the estimate from H&FvB. Thus, there are relatively large scaling factors for the production ratio relative to Holocene for our calculated Pinedale and Bull Lake fluxes (0.71 and 0.80), as well as for Graly et al. 2011 (0.67), because the solar modulation is considerably different from Modern (500 MV, using Steinhilber as a modern common reference) to the Holocene-average (280 MV). It is largest for the Graly estimate because we had to scale from 700 MV to 500 MV first, as described above. We finally multiply the calculated or estimated flux by the reciprocal of the production ratio relative to the Holocene*

*Please note that there was previously a typo/drafting error for this value for Graly et al. (2011) in column 5 of Table 3. It should be 0.67 (the value used in all calculations). This has been fixed.*

*We have now added a helpful flow-chart to the Supplemental Material (Fig. S1) that walks the reader through these calculations with a worked example. This will also be helpful to other workers who may need to do paleomagnetic normalizations such as this in the future.*

**295: This is not true. A linear fit is significantly better for the Bull Lake data (if you exclude the inheritance-dominated samples).**

*If we were to exclude all samples beneath 60 cm for Bull Lake, a second or third order polynomial is an even better fit -- however, we are referring to the measured depth profile as a whole. We have an expectation of exponential decline for these soil profiles from reactive transport modeling (Maher & von Blanckenburg, 2016) and when considering these profiles wholly we observe this. It's the simplest explanation.*

**308: Arguably, the bulge may have been missed in the Bull Lake profile, as no sample at equivalent depth was analyzed.**

*Certainly possible. We make note of this in the text now in this section as follows:*

> *It is possible that such an increase may have been missed in the Bull Lake profile, as the equivalent 10cm depth interval was not sampled.*

**311-323: I would still like to see this subject treated in more depth. And, I still think that a preference for larger grain sizes in mixing (not downward transport, but mixing) is the most logical explanation.**

*We have revised the sentence on line 314 and added text to note that while this is a possibility, we don't have enough information to further assess its validity. The text now reads:*

> "The different grain sizes analyzed here and in Schaller et al. (2009a) might exhibit different diffusion coefficients, by which larger grain sizes are more easily mixed, however a trend in smaller grain size fractions with depth within the $^{10}Be_{in\ situ}$ mixing layer would likely be observed if this were the case. Unfortunately, separate grain size classes were not measured for $^{10}Be_{in\ situ}$ within the full mixing zone of either profile to further assess this explanation."

**Line 328: Since you linearly sampled at random from something that varies on an exponential scale, this result is expected. (Though as I mentioned above, I believe this whole term to be in error.)**

*Given our new error propagation uncertainty treatment, we have removed this sentence describing Monte Carlo simulations. The text now reads:*

> "While the possibility of desorption cannot be ruled out, it's unlikely that either profile has experienced loss to such a degree, as pH, and thus Kd and retentivity, increases with depth. Even in the worst-case scenario of assuming maximum possible loss at the lowest Kd estimate, the magnitude of the potential loss does not substantially affect our calculated $F_{(10Bemet)}$ estimates within uncertainties. Our calculations thus capture the potential maximal bound for loss via propagated uncertainties."

**Line 345: I don't know why you took out the +20% paleo-precipitation from (Birkel et al., 2012). I thought that was a very useful point to bring in. My previous review stated only that you could not meaningfully use it as an upper bound when comparing the Graly et al. value to your results. Quoting the highest regional precipitation rate seems far less useful a fact than a paleo-precipitation model result.**

*We quote a higher precipitation rate not to place an upper bound on the Graly et al. (2011) estimate for this local site, but to highlight that within the same region/cell covered by the Heikkila and von Blanckenburg (2015) estimate, substantial local differences in average precipitation exist. This leads to estimates that considerably differ for localities using Graly (local precip. input) compared to H&FvB (regional). Within the area contained by the grid cell, the former estimate can scale considerably based on precipitation input, while the latter will always be the same. This is an important point that future workers should consider when estimating meteoric $^{10}Be$ fluxes, particularly in study areas nearby mountain ranges/precipitation gradients. We make this clearer by adding to the text as follows:*

> For example, if one were to estimate $F_{(10Bemet)}$ from Graly et al. (2011) via (Eq. 1) to nearby Fish Lake Mountain contained within the same Heikkilä and von Blanckenburg (2015) grid cell as this study site, with a modern precipitation rate of 128 cm y$^{-1}$ (WRCC, 2005), the $F_{(10Bemet)}$ would be 2.5 x 10$^6$ atoms cm$^{-2}$ y$^{-1}$, substantially higher than that predicted from Heikkilä and von Blanckenburg (2015).

*The +20% paleo-precipitation rate was previously applied directly to our calculations, which is overly credulous to Birkel et al. (2012)'s coarse-resolution modeled results that might not be applicable to this low elevation site in the Wind River mountains. This is a separate discussion point -- that paleo-precipitation may have been higher at this particular site does not serve to illustrate that the region covered by a grid cell of Heikkila and von Blanckenburg (2015) is large and can include areas with substantial precipitation gradients that would give rise to different flux estimates between methods (as we see here).*

**Table 1: The final column should have inheritance subtracted.**

*Agreed, this has been fixed both here and in the reporting of calculated inventory values in section 4.1.*

**Table 3: Uncertainties must be included for the Graly et al. (2011) line of this table. A root mean square error is provided in the publication.**

*Indeed! This has been fixed, thanks for pointing it out.*

**I have no idea how the 0.83 value for the Graly et al. (2011) Holocene F term is derived. It is certainly not the first column divided by the third column, as the others seem to be. I am equally mystified as to how the Holocene correction factor for the Graly line is derived.**

*There is an unfortunate typo/drafting error here. The correction factor relative to Holocene should be 0.665 (rounded to 0.67), not 1.06. The typo value of 1.06 was never used in any calculation. This has been fixed. This scaling factor is calculated by dividing the Graly scaling factor relative to Modern (0.82) by the Heikkila scaling factor relative to Modern (1.23) for a value of 0.67. Then, we divide the calculated flux (0.55) by this value (0.67) to arrive at a rescaled flux of 0.83E6 at/cm$^2$/y.*

**The Heikkilä line must also have uncertainties. These are provided in the publication. The industrial run is not an uncertainty. It is a different result.**

*We previously phrased this poorly in the footnote -- this is now changed (thanks for pointing this out):*

> [d] *uncertainty represents the difference between the 'industrial' and the "pre-industrial" modeled flux of Heikkilä and von Blanckenburg (2015)*

**Reviewer 2**

**I just have a few suggestions for minor edits prior to publication.**

**On line 87, the grain size fraction should be <2 um.**

*Fixed, thank you.*

**In the results section, the authors should refer to the equations they used to calculate the inventory values.**

*Done.*

**Table 2 is the first table reference in the text, so it should probably be renumbered to Table 1. Alternatively, this table could go into the supplementary material.**

*Good catch, this has been changed.*

[revised manuscript text omitted]